# FRMD8 promotes inflammatory and growth factor signalling by stabilising the iRhom/ADAM17 sheddase complex

Ulrike Künzel, Adam Graham Grieve, Yao Meng†, Boris Sieber, Sally A Cowley, Matthew Freeman*

Sir William Dunn School of Pathology, University of Oxford, Oxford, United Kingdom

**Abstract** Many intercellular signals are synthesised as transmembrane precursors that are released by proteolytic cleavage ('shedding') from the cell surface. ADAM17, a membrane-tethered metalloprotease, is the primary shedding enzyme responsible for the release of the inflammatory cytokine TNFα and several EGF receptor ligands. ADAM17 exists in complex with the rhomboid-like iRhom proteins, which act as cofactors that regulate ADAM17 substrate shedding. Here we report that the poorly characterised FERM domain-containing protein FRMD8 is a new component of the iRhom2/ADAM17 sheddase complex. FRMD8 binds to the cytoplasmic N-terminus of iRhoms and is necessary to stabilise iRhoms and ADAM17 at the cell surface. In the absence of FRMD8, iRhom2 and ADAM17 are degraded via the endolysosomal pathway, resulting in the reduction of ADAM17-mediated shedding. We have confirmed the pathophysiological significance of FRMD8 in iPSC-derived human macrophages and mouse tissues, thus demonstrating its role in the regulated release of multiple cytokine and growth factor signals.
DOI: https://doi.org/10.7554/eLife.35012.001

*For correspondence:
matthew.freeman@path.ox.ac.uk

Present address: †Department of Biochemistry, University of Oxford, Oxford, United Kingdom

## Introduction

The cell surface protease ADAM17 (also called TACE) mediates the release of many important signalling molecules by 'shedding' their extracellular ligand domains from transmembrane precursors. A prominent example is the role of ADAM17 in releasing tumour necrosis factor alpha (TNFα) (*Black et al., 1997*; *Moss et al., 1997*), a primary cytokine involved in the inflammatory responses to infection and tissue damage (*Kalliolias and Ivashkiv, 2016*). In addition, ADAM17 is the principal sheddase of the epidermal growth factor (EGF) receptor ligands amphiregulin (AREG), transforming growth factor alpha (TGFα), heparin-binding EGF (HB-EGF), epigen, and epiregulin (*Sahin et al., 2004*; *Sahin and Blobel, 2007*). The control of ADAM17 activity has therefore been the focus of much fundamental and pharmaceutical research (reviewed in [*Rose-John, 2013*; *Zunke and Rose-John, 2017*]). We and others have previously reported that the rhomboid-like iRhom proteins have a specific and extensive regulatory relationship with ADAM17, to the extent that iRhoms can effectively be considered as regulatory subunits of the protease (*Grieve et al., 2017*). iRhoms are members of a wider family of evolutionarily related multi-pass membrane proteins, called the rhomboid-like superfamily (*Freeman, 2014*). The family is named after the rhomboids, intramembrane serine proteases that cleave substrate transmembrane domains (TMDs), but many members, including iRhoms, have lost protease activity during evolution. iRhom1 and its paralogue iRhom2 (encoded by the genes *RHBDF1* and *RHBDF2*) show redundancy in regulating ADAM17 maturation, but differ in their tissue expression (*Christova et al., 2013*). Many cell types express both iRhoms, so the loss of one can be compensated by the other (*Christova et al., 2013*; *Li et al., 2015*). Macrophages are, however, an exception: iRhom1 is not expressed, so iRhom2 alone regulates ADAM17 and therefore

**eLife digest** Cells in the human body communicate with one another for many different reasons, including to help organs develop correctly and to produce a healthy reponse to injury and infection. Signalling proteins, such as growth factors and cytokines, form the main language of this communication.

Initially, many growth factors and cytokines remain attached to the surface of the cell that made them. When cells need to send a message to another one, an enzyme called ADAM17 acts like a pair of scissors to release the proteins from the cell surface, allowing them to travel towards other cells. This process must be carefully controlled because releasing too many growth factors or cytokines (or releasing them at inappropriate times) can lead to cancer and inflammatory diseases such as rheumatoid arthritis.

Another group of proteins called iRhoms bind to ADAM17 to regulate the enzyme's activity. But what controls the activity of the iRhom proteins themselves? To find out, Künzel et al. used a technique called a proteomic screen that can identify which proteins bind to each other. This revealed that a protein called FRMD8 binds to iRhoms. Further experiments in human cells and mice revealed that FRMD8 maintains adequate levels of both ADAM17 and iRhoms at the surface of the cell. Cells that lack FRMD8 break down ADAM17 and iRhom proteins and release fewer growth factors and cytokines.

Further work could help us to learn whether stopping FRMD8 from interacting with iRhoms could reduce cell communication. This, in turn, might reduce inflammation or cell growth. If so, then developing drugs that prevent FRMD8 from binding to iRhoms could lead to new treatments for inflammatory diseases and cancer.

DOI: https://doi.org/10.7554/eLife.35012.002

TNFα inflammatory signalling in macrophages (*Adrain et al., 2012*; *McIlwain et al., 2012*; *Issuree et al., 2013*). iRhoms control ADAM17 activity in multiple ways. First, they bind to the catalytically immature pro-form of ADAM17 (proADAM17) in the endoplasmic reticulum (ER), and are required for its trafficking from the ER to the Golgi apparatus (*Adrain et al., 2012*; *McIlwain et al., 2012*). Once proADAM17 reaches the Golgi, it is matured by the removal of its inhibitory pro-domain by pro-protein convertases (*Schlöndorff et al., 2000*; *Endres et al., 2003*) and is further trafficked to the plasma membrane. iRhoms have further regulatory functions beyond this step of ADAM17 maturation. Still bound to each other, iRhom2 prevents the lysosomal degradation of ADAM17 (*Grieve et al., 2017*). Later, iRhom2 controls the activation of ADAM17: the phosphorylation of the iRhom2 cytoplasmic tail promotes the recruitment of 14-3-3 proteins, which promote the shedding activity of ADAM17, thereby releasing TNFα from the cell surface in response to inflammatory triggers (*Grieve et al., 2017*; *Cavadas et al., 2017*). Finally, iRhoms are also reported to contribute to ADAM17 substrate specificity (*Maretzky et al., 2013*). This intimate regulatory role of iRhoms make them essential players in ADAM17-mediated signalling and thus new targets for manipulating inflammatory signalling. The significance of this potential is underlined by the fact that anti-TNFα therapies, used to treat rheumatoid arthritis and other inflammatory diseases, are currently the biggest grossing drugs in the world (*Monaco et al., 2015*).

Despite the role of the iRhom/ADAM17 shedding complex in controlling signalling, much is yet to be understood about the molecular mechanisms that control this inflammatory trigger. To identify the wider machinery by which iRhoms regulate ADAM17, we report here a proteomic screen to identify their binding partners. We have identified the poorly characterised FERM domain-containing protein 8 (FRMD8) as having a strong and specific interaction with the cytoplasmic N-terminus of iRhoms. The functional significance of this interaction is demonstrated by loss of FRMD8 causing a similar phenotype to iRhom deficiency in cells: loss of mature ADAM17 and severely reduced shedding of ADAM17 substrates from the cell surface. We show that loss of FRMD8 leads to lysosomal degradation of mature ADAM17 and iRhom2, indicating that its function is to stabilise the iRhom/ADAM17 sheddase complex once it reaches the plasma membrane. Overall, our results imply that FRMD8 is an essential component of the inflammatory signalling machinery. To test this proposal *in vivo* we deleted the *FRMD8* gene in human induced pluripotent stem cells (iPSCs) and differentiated

them into macrophages. Consistent with our biochemical data, these mutant macrophages were defective in their ability to release TNFα in response to lipopolysaccharide (LPS) stimulation, demonstrating the pathophysiological importance of FRMD8 in the normal inflammatory response by human macrophages. The *in vivo* significance of FRMD8 in regulating the stability of the iRhom/ADAM17 shedding complex was further reinforced by our observation that mature ADAM17 and iRhom2 protein levels are strongly reduced in tissues of FRMD8-deficent mice.

## Results

### FRMD8 is a novel interaction partner of iRhom1 and iRhom2

To investigate the molecular mechanisms that underlie iRhom2 functions, we performed a mass spectrometry-based screen to identify new proteins that interact with human iRhom2. iRhom2-3xHA was stably expressed in human embryonic kidney (HEK) 293T cells and immunoprecipitated. The bead eluates containing immunoprecipitated iRhom2 and its interacting proteins were analysed by label-free mass spectrometry. As a negative control, we did the same analysis in parallel with 3xHA-tagged UNC93B1, an unrelated polytopic protein that, like iRhom2, is predominantly located in the ER (*Koehn et al., 2007*) (*Figure 1—figure supplement 1A*). Quantitative protein abundance data from three biological replicates of iRhom2 and UNC93B1 co-immunoprecipitations were statistically analysed using the Perseus software platform (*Tyanova et al., 2016*). Validating the overall approach, we detected ADAM17, the known iRhom2 interacting protein (*Adrain et al., 2012*; *McIlwain et al., 2012*; *Christova et al., 2013*) as a statistically significant hit (*Figure 1A*, *Table 1*). Among the hits were several 14-3-3 proteins (eta, epsilon, gamma, sigma, theta, zeta/delta) and MAPK1/3 (*Table 1*), which we have previously reported to participate in the regulation of inflammatory signalling by phosphorylation of iRhom2 (*Grieve et al., 2017*). The top hit by a long way, however, was FRMD8 (*Figure 1A*, *Table 1*), a poorly studied protein that has not previously been implicated in iRhom function, ADAM17 regulation, and growth factor or cytokine signalling.

We confirmed the interaction between iRhom2 and FRMD8 by immunoprecipitation. C-terminally V5-tagged FRMD8 co-immunoprecipitated with either iRhom1-3xHA or iRhom2-3xHA (*Figure 1B*). Conversely, we pulled down both iRhom1-3xHA and iRhom2-3xHA with an antibody against the V5 tag. Finally, we were also able to co-immunoprecipitate endogenous FRMD8 with iRhom2-3xHA (*Figure 1—figure supplement 1B*). Together these results identify FRMD8 as a bona fide binding partner of iRhom1 and iRhom2 in human cells.

### FRMD8 is required for iRhom function

As its name indicates, FRMD8 is a FERM (4.1/ezrin/radixin/moesin) domain-containing protein. It is predicted to be a soluble cytoplasmic protein, and the only report about its function describes it as binding to the Wnt accessory receptor low-density lipoprotein receptor-related protein 6 (LRP6), and negatively regulating Wnt signalling (*Kategaya et al., 2009*). To investigate the functional significance of FRMD8 binding to iRhoms, we examined the effects of loss of FRMD8 on iRhom function in HEK293T cells, using both siRNA and CRISPR/Cas9-mediated gene deletion (*Figure 2A,B*). In both cases, loss of FRMD8 drastically reduced the protein levels of mature ADAM17 (*Figure 2A,B*). This effect was specific to ADAM17, as the maturation of its closest homologue, ADAM10, was unaffected by loss of FRMD8 (*Figure 2B*). Moreover, mature ADAM17 levels were rescued by expression of FRMD8-V5 in FRMD8 knockout HEK293T cells (*Figure 2C*), confirming that the phenotype was caused by FRMD8 loss. Finally, in addition to this reduction of mature ADAM17 caused by FRMD8 loss, we found a striking loss of ADAM17, but not ADAM10, on the cell surface (*Figure 2D*). These phenotypes partially phenocopy the loss of iRhoms (*Christova et al., 2013*; *Grieve et al., 2017*), consistent with FRMD8 being needed for iRhoms to act as positive regulators of ADAM17.

We also examined the consequences of loss of FRMD8 on ADAM17-dependent signalling. The shedding of alkaline phosphatase (AP)-tagged EGF receptor ligands AREG and HB-EGF, after stimulation with phorbol 12-myristate 13-acetate (PMA), were both substantially reduced in FRMD8 knockout cells (*Figure 2E*). To exclude the possibility that the defect in FRMD8 knockout cells is an inability to respond to PMA, we measured both PMA-stimulated and unstimulated, constitutive shedding of AP-tagged TGFα, another major EGFR ligand. Again, FRMD8 knockout cells released significantly less AP-TGFα compared to wild-type cells, both after stimulation but also after 20 hr of

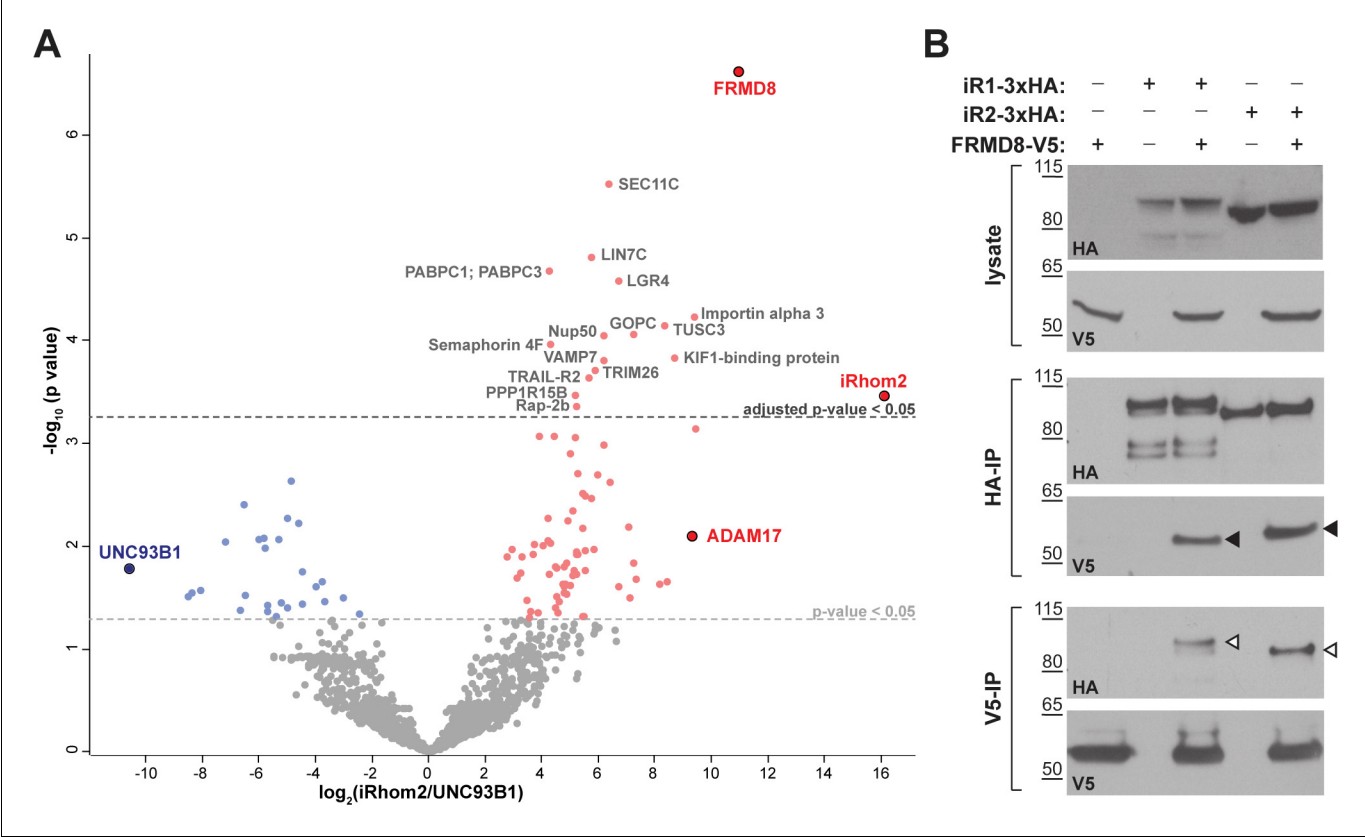

**Figure 1.** FRMD8 is a novel interaction partner of iRhom1 and iRhom2. (**A**) Volcano plot representing results from three iRhom2 co-immunoprecipitations. The fold change of label-free quantification values (in log2 ratio) was plotted against the p value (-log10 transformed). The grey dotted line indicates p-values <0.05 (analysed with a two-sample t-test). Benjamini-Hochberg correction was applied to adjust the p-value for multiple hypothesis testing (dark grey dotted line). (**B**) Lysates of HEK293T cells stably expressing human iRhom1-3xHA or iRhom2-3xHA transfected with human FRMD8-V5 (where indicated) were subjected to anti-HA and anti-V5 immunoprecipitation (HA-IP, V5–IP) and a western blot using anti-HA and anti-V5 antibodies was performed. Black arrowheads indicated the co-immunoprecipitated FRMD8-V5; white arrowheads indicated the co-immunoprecipitated iRhoms.

DOI: https://doi.org/10.7554/eLife.35012.003

The following figure supplement is available for figure 1:

**Figure supplement 1.** Setup and confirmation of the mass spectrometry screen.

DOI: https://doi.org/10.7554/eLife.35012.004

constitutive shedding (*Figure 2F*), implying that mutant cells had fundamental defects in their ability to shed ADAM17 ligands, regardless of PMA stimulation. To demonstrate that the release of ligands was indeed caused by metalloprotease shedding and not simply an indication of leakage caused by cell death, we showed that it was sensitive to the ADAM10/17 inhibitor GW280264X (GW) (*Figure 2E,F*). Overall, as with ADAM17 maturation, the shedding defects in FRMD8-deficient cells resemble those caused by the loss of iRhoms.

## FRMD8 binds to the cytoplasmic iRhom2 N-terminus and regulates mature ADAM17 levels

As described above, iRhoms regulate ADAM17 function at multiple stages: from ER-to-Golgi trafficking, to the activation of the sheddase at the cell surface. To address where FRMD8 fits in this long-term relationship between iRhoms and ADAM17, we started by analysing the FRMD8 binding site within iRhom2. As a cytoplasmic protein, FRMD8 was likely to bind to the only substantial cytoplasmic region of iRhom2, its N-terminus. We therefore made a set of iRhom2-3xHA N-terminal deletion constructs (*Figure 3A*) to locate the binding site. Deletion of the first 200 amino acids in the N-terminus of iRhom2 (iRhom2$^{\Delta200}$) did not disrupt FRMD8 binding, but no interaction was

**Table 1.** List of iRhom2 interaction partners identified in the mass spectrometry screen that have either shown a significant adjusted p-value or been reported previously (**Adrain et al., 2012**; **McIlwain et al., 2012**; **Grieve et al., 2017**).

P-values from a two-sample t-test in Perseus are listed below. P-values were adjusted for multiple hypothesis testing with the Benjamini-Hochberg correction and are listed under 'adjusted p-values'.

| Prot. ID | Protein name | Gene | p-value | Adjusted p-value |
|---|---|---|---|---|
| Q9BZ67 | FERM domain-containing protein 8 | FRMD8 | $2.44 \cdot 10^{-7}$ | $2.38 \cdot 10^{-4}$ |
| Q9BY50 | Signal peptidase subunit SEC11C | SEC11C | $2.94 \cdot 10^{-6}$ | $1.71 \cdot 10^{-3}$ |
| Q9NUP9 | Protein lin-7 homolog C | LIN7C | $1.55 \cdot 10^{-5}$ | $6.45 \cdot 10^{-3}$ |
| P11940 | Polyadenylate-binding protein 1; Polyadenylate-binding protein 3 | PABPC1; PABPC3 | $2.10 \cdot 10^{-5}$ | $7.66 \cdot 10^{-3}$ |
| Q9BXB1 | Leucine-rich repeat-containing GPCR 4 | LGR4 | $2.63 \cdot 10^{-5}$ | $8.53 \cdot 10^{-3}$ |
| O00629 | Importin subunit alpha-3 | KPNA4 | $5.76 \cdot 10^{-5}$ | $1.40 \cdot 10^{-2}$ |
| Q13454 | Tumor suppressor candidate 3 | TUSC3 | $7.05 \cdot 10^{-5}$ | $1.58 \cdot 10^{-2}$ |
| Q9HD26 | GOPC/PIST | GOPC | $8.54 \cdot 10^{-5}$ | $1.78 \cdot 10^{-2}$ |
| Q9UKX7 | Nuclear pore complex protein Nup50 | NUP50 | $8.77 \cdot 10^{-5}$ | $1.71 \cdot 10^{-2}$ |
| O95754 | Semaphorin-4F | SEMA4F | $1.06 \cdot 10^{-4}$ | $1.83 \cdot 10^{-2}$ |
| Q96EK5 | KIF1-binding protein | KIAA1279 | $1.47 \cdot 10^{-4}$ | $2.26 \cdot 10^{-2}$ |
| P51809 | Vesicle-associated membrane protein 7 | VAMP7 | $1.55 \cdot 10^{-4}$ | $2.26 \cdot 10^{-2}$ |
| Q12899 | Tripartite motif-containing protein 26 | TRIM26 | $1.93 \cdot 10^{-4}$ | $2.68 \cdot 10^{-2}$ |
| O14763 | TRAIL receptor 2 | TNFRSF10B | $2.29 \cdot 10^{-4}$ | $3.04 \cdot 10^{-2}$ |
| Q5SWA1 | Protein phosphatase 1 subunit 15B | PPP1R15B | $3.35 \cdot 10^{-4}$ | $4.25 \cdot 10^{-2}$ |
| Q6PJF5 | iRhom2 | RHBDF2 | $3.42 \cdot 10^{-4}$ | $4.16 \cdot 10^{-2}$ |
| P61225 | Ras-related protein Rap-2b | RAP2B | $4.27 \cdot 10^{-4}$ | $4.98 \cdot 10^{-2}$ |
| P28482 | Mitogen-activated protein kinase 1 | MAPK1 | $3.37 \cdot 10^{-3}$ | 0.22 |
| P27361 | Mitogen-activated protein kinase 3 | MAPK3 | $5.34 \cdot 10^{-3}$ | 0.32 |
| P62258 | 14-3-3 protein epsilon | YWHAE | $6.61 \cdot 10^{-3}$ | 0.35 |
| P78536 | ADAM17 | ADAM17 | $8.07 \cdot 10^{-3}$ | 0.40 |
| P63104 | 14-3-3 protein zeta/delta | YWHAZ | $9.14 \cdot 10^{-3}$ | 0.41 |
| P27348 | 14-3-3 protein theta | YWHAQ | $1.20 \cdot 10^{-2}$ | 0.45 |
| P31947 | 14-3-3 protein sigma | SFN | $2.19 \cdot 10^{-2}$ | 0.63 |
| Q04917 | 14-3-3 protein eta | YWHAH | $2.33 \cdot 10^{-2}$ | 0.65 |
| P61981 | 14-3-3 protein gamma | YWHAG | $3.15 \cdot 10^{-2}$ | 0.75 |
| P31946 | 14-3-3 protein alpha/beta | YWHAB | $6.53 \cdot 10^{-2}$ | 1 |
| P51812 | Ribosomal protein S6 kinase alpha-3 | RPS6KA3 | $6.53 \cdot 10^{-2}$ | 1 |

DOI: https://doi.org/10.7554/eLife.35012.005

detected in mutants greater than Δ300 (**Figure 3B**), implying that the region between 200 and 300 amino acids was necessary for FRMD8 binding. An internal deletion of amino acids 201–300 within iRhom2 (iRhom2$^{\Delta201-300}$) led to the loss of exogenous FRMD8 binding (**Figure 3C**), confirming that the FRMD8 binding site lies within this region. In line with this, an iRhom1/2 DKO cell line reconstituted with iRhom2$^{\Delta201-300}$ showed a similar deficiency to FRMD8 KO cells in ADAM17-mediated shedding of AREG (compare **Figures 3D** and **2E**). This reduction in shedding correlates with a reduction in the level of mature ADAM17 (**Figure 3E**). Overall, this makes the iRhom2$^{\Delta201-300}$ mutant a useful tool to study the loss of FRMD8 binding to iRhom2 and highlights that FRMD8 binding affects levels of mature ADAM17, presumably either through controlling ADAM17 maturation or stability. Interestingly, the FRMD8 binding site is also absent in a mouse iRhom2 mutant called *curly-bare (cub)*, which lacks residues 1–268 (**Hosur et al., 2014**; **Siggs et al., 2014**). Sequence alignment

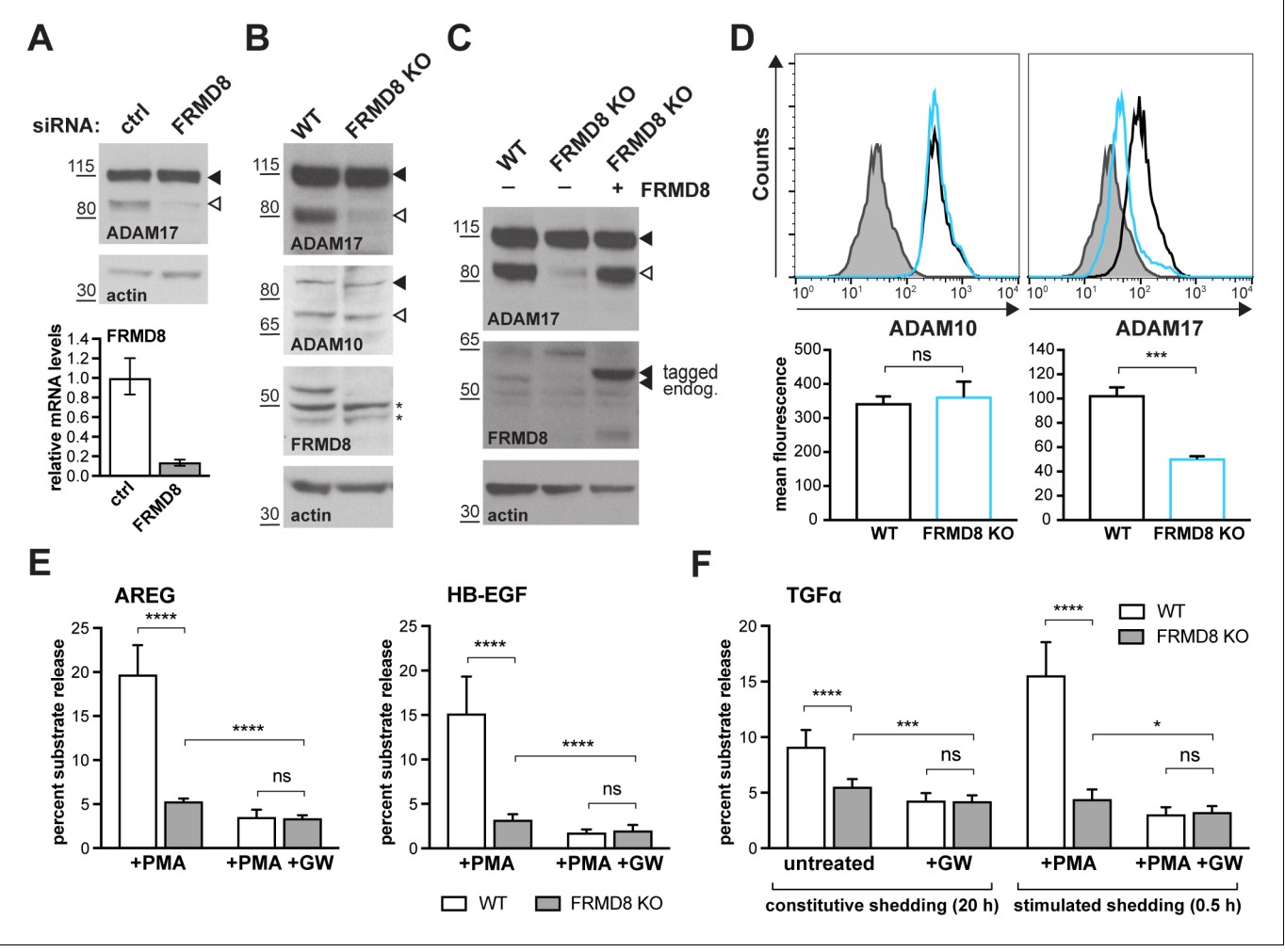

**Figure 2.** FRMD8 loss reduces mature ADAM17 levels and impairs ADAM17-dependent shedding activity. (A) ADAM17 levels were analysed in HEK293T cells transfected with non-targeting siRNA control pool (ctrl) or FRMD8 SMARTpool siRNA after western blotting with anti-ADAM17 and anti-actin staining. In this and subsequent figures, pro- and mature form of ADAM17 are indicated with black and white arrowheads, respectively. Lower panel: Knockdown efficiency of FRMD8 was analysed by TaqMan PCR. (B, C) Lysates from wild-type (WT) and FRMD8 knockout (KO) HEK293T cells, transiently transfected with FRMD8-V5 for 72 hr (where indicated) and immunoblotted for endogenous ADAM17, ADAM10, FRMD8 and actin using western blotting. Nonspecific bands are marked with an asterisk. (D) Cell surface levels of endogenous ADAM10 and ADAM17 were analysed in WT and FRMD8 KO HEK293T cells after stimulation with 200 nM PMA for 5 min. Unpermeabilised cells were stained on ice with ADAM10 and ADAM17 antibodies, or only with the secondary antibody as a control (grey). The immunostaining was analysed by flow cytometry. The graph shown is one representative experiment out of four biological replicates. The geometric mean fluorescence was calculated for each experiment using FlowJo software. Statistical analysis was performed using an unpaired t-test. (E, F) WT and FRMD8 KO HEK293T cells were transiently transfected with alkaline phosphatase (AP)-tagged AREG, HB-EGF or TGFα, and then either incubated with 200 nM PMA, with 200 nM PMA and 1 µM GW (ADAM10/ADAM17 inhibitor), or with DMSO for 30 min. In addition, cells transfected with AP-TGFα were either left unstimulated for 20 hr or incubated with GW for 20 hr. AP activity was measured in supernatants and cell lysates. Each experiment was performed in biological triplicates. The results of three independent shedding experiments are shown. Statistical analysis was performed of using a Mann-Whitney test. ns = p value>0.05; *=p value<0.05; ***=p value<0.001; ****=p value<0.0001.

DOI: https://doi.org/10.7554/eLife.35012.006

shows that the deletion of 268 amino acids in mouse iRhom2 corresponds to the loss of residues 1–298 in the human protein (*Figure 3—figure supplement 1A*). Consistent with this mapping data, we found that whereas full-length mouse iRhom2 bound human FRMD8, the *cub* mutant cannot (*Figure 3—figure supplement 1B*). This failure of FRMD8 binding presumably contributes to the complex defects that underlie the *cub* phenotype (*Johnson et al., 2003*; *Hosur et al., 2014*; *Siggs et al., 2014*).

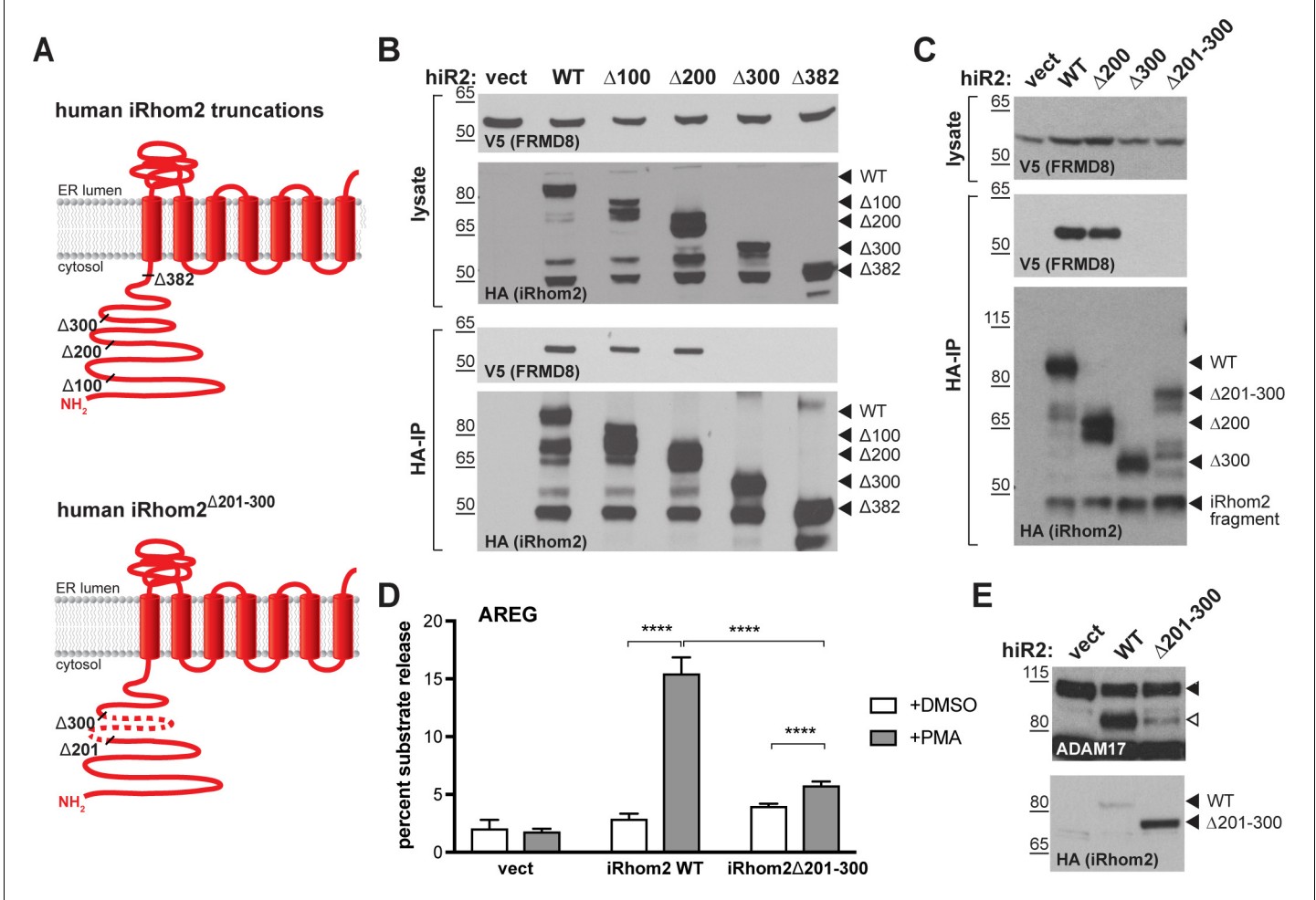

**Figure 3.** FRMD8 binds to the iRhom2 N-terminus. (A) Schematic representation of truncated human iRhom2 constructs used in (B–E). (B, C) Lysates and anti-HA immunoprecipitation (HA-IP) from HEK293T cells transiently co-transfected with FRMD8-V5 and either empty vector (vect) or truncated human iRhom2-3xHA constructs were immunoblotted for V5 and HA. (D) iRhom1/2 double knockout HEK293T cells stably expressing empty vector (vect) or human iRhom2-3xHA constructs were transiently transfected with alkaline phosphatase (AP)-tagged AREG and then incubated with 200 nM PMA or with DMSO for 30 min. AP activity was measured in supernatants and cell lysates. Each experiment was performed in biological triplicates. The results of three independent shedding experiments are shown. Statistical analysis was performed using a Mann-Whitney test. ****=p value<0.0001. (E) Lysates from iRhom1/2 double knockout HEK293T cells transiently transfected with empty vector (vect) or human iRhom2-3xHA constructs were immunoblotted for ADAM17 and HA.

DOI: https://doi.org/10.7554/eLife.35012.007

The following figure supplement is available for figure 3:

**Figure supplement 1.** A) Amino acid sequence alignment of human and mouse iRhom2 N-terminal region using Clustal Omega.

DOI: https://doi.org/10.7554/eLife.35012.008

Combined, these data show that FRMD8 is recruited to a discrete 201–300 amino acid region of the iRhom2 N-terminus, and that this binding is required for sufficient levels of mature ADAM17, as well as ADAM17-dependent shedding.

## FRMD8, iRhom2, and ADAM17 form a tripartite complex

We next investigated the nature of a putative tripartite complex between iRhom2, FRMD8 and ADAM17. Previous work has shown a strong interaction between iRhom2 and ADAM17 (*Adrain et al., 2012*; *McIlwain et al., 2012*; *Grieve et al., 2017*), yet how FRMD8 intersects with this complex is not known. When performing an immunoprecipitation of FRMD8, we found both immature and mature ADAM17 as well as iRhom2 (*Figure 4A*). This indicates that FRMD8 does

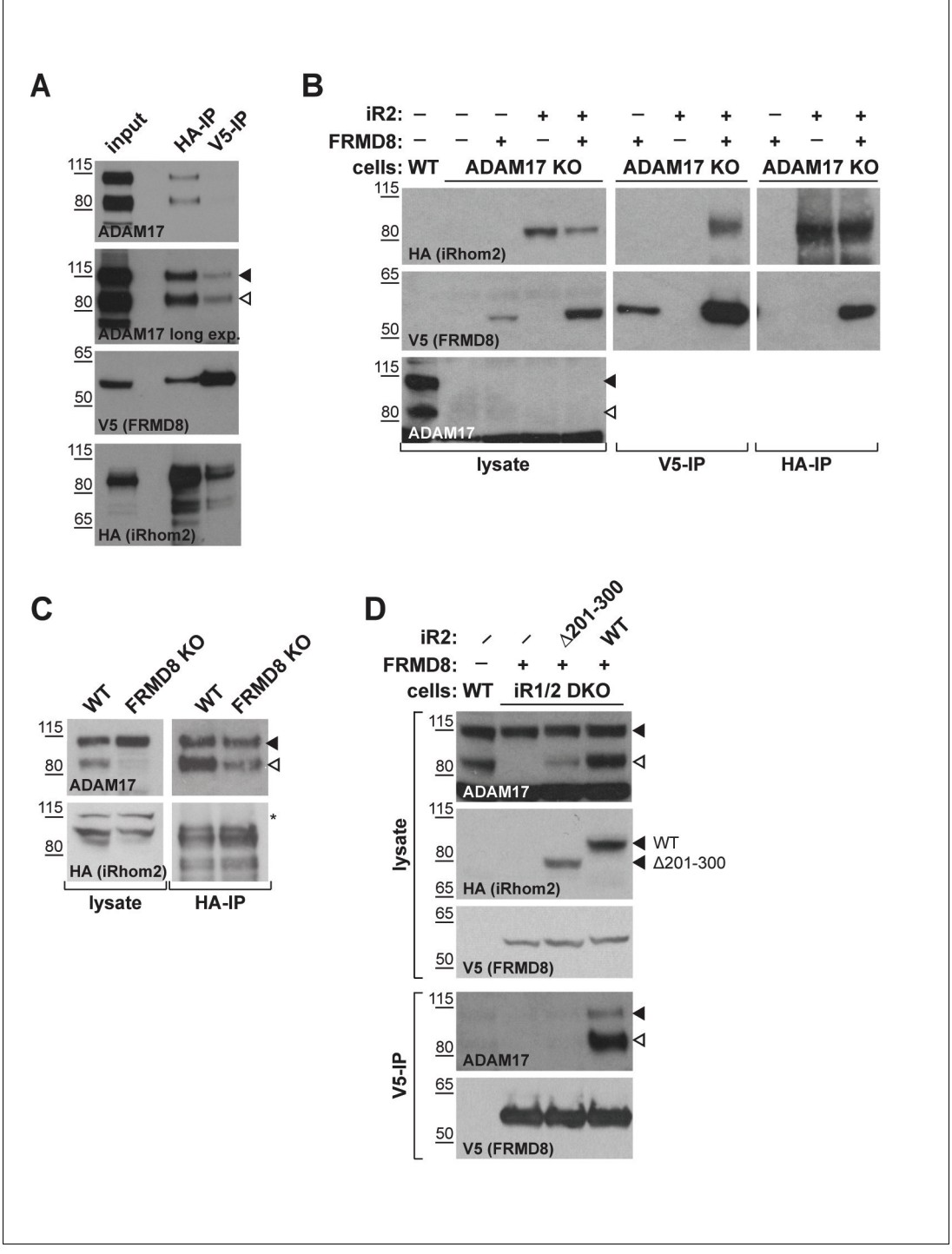

**Figure 4.** iRhom2 binds to FRMD8 and ADAM17 simultaneously. (**A**) Lysates, anti-HA and anti-V5 immunoprecipitations (HA-IP, V5–IP) of HEK293T cells co-expressing human iRhom2-3xHA and human FRMD8-V5 were immunoblotted for ADAM17, HA and V5. (**B**) Lysates of wild-type (WT) and ADAM17 knockout (KO) HEK293T cells were transiently transfected with human iRhom2-3xHA and FRMD8-V5 (where indicated), anti-HA and anti-V5 immunoprecipitated (HA-IP; V5–IP) and immunoblotted for ADAM17, HA, and V5. (**C**) Lysates of WT and FRMD8 KO HEK293T cells stably expressing human iRhom2-3xHA were anti-HA immunoprecipitated (HA-IP) and stained for ADAM17 and HA. Nonspecific bands are indicated by an asterisk. (**D**) Lysates of WT and iRhom1/2 double knockout (DKO) HEK293T cells stably expressing human iRhom2$^{WT}$-3xHA or iRhom2$^{\Delta201-300}$-3xHA were anti-V5 immunoprecipitated (V5–IP) and immunoblotted for ADAM17, HA and V5.
DOI: https://doi.org/10.7554/eLife.35012.009

indeed form a tripartite complex with ADAM17 and iRhom2. To test this further, we performed a series of pairwise co-immunoprecipitations in the absence of FRMD8, ADAM17 or iRhoms. First we tested the requirement for ADAM17 in the iRhom2/FRMD8 interaction. We found that exogenous FRMD8 and iRhom2 co-immunoprecipitated with each other in ADAM17 knockout cells (*Figure 4B*). In turn, iRhom2 and ADAM17 were still able to interact with each other in FRMD8 knockout cells (*Figure 4C*), showing that FRMD8 is not essential for the iRhom/ADAM17 sheddase complex to form. In contrast, FRMD8 did not pull down ADAM17 in cells mutant for both iRhoms (*Figure 4D*). This demonstrated that there is no direct link between FRMD8 and ADAM17; instead both bind simultaneously to iRhom2. Supporting this, FRMD8 co-immunoprecipitated with pro- or mature ADAM17 in iRhom1/2 DKO cells reconstituted with iRhom2$^{WT}$, but not with iRhom2$^{\Delta201-300}$ (*Figure 4D*), the mutant that does not bind to FRMD8 (*Figure 3C*). FRMD8 binds to both iRhom2/proADAM17 and iRhom2/mature ADAM17 complexes but associates preferentially with iRhom2/mature ADAM17 complexes (*Figure 4A,D*), which have been shown to exist at the cell surface (*Grieve et al., 2017*; *Cavadas et al., 2017*). This is consistent with the observation of specific effects of FRMD8 loss on mature ADAM17 at the cell surface, but not immature ADAM17 (*Figure 2A–C*).

## FRMD8 recruitment promotes cell surface localisation of iRhom2 and ADAM17

To further investigate a potential role for FRMD8 at the cell surface, we first assessed its effects on iRhom2 localisation by immunofluorescence. Overexpression of FRMD8-V5 in iRhom1/2 DKO cells reconstituted with wild-type iRhom2 led to a striking increase in plasma membrane iRhom2 (*Figure 5A*), which in wild-type cells is almost exclusively observed within the endoplasmic reticulum (*Figure 1—figure supplement 1A*). As a control, the iRhom2$^{\Delta300}$ mutant, which cannot bind to FRMD8 (*Figure 3B*), did not undergo the same ER-to-plasma membrane relocalisation upon FRMD8 overexpression. Indicating a reciprocal relationship between the two proteins, we also observed that the iRhom2 N-terminus was required for FRMD8 localisation at the cell surface (*Figure 5A,B*).

To test whether FRMD8 is sufficient to target iRhom2 to the cell surface, we fused FRMD8 to the N-terminus of the ER-localised iRhom2$^{\Delta300}$ mutant (FRMD8-iRhom2$^{\Delta300}$). Strikingly, we saw that the normal ER localisation of iRhom2$^{\Delta300}$ (*Figure 5D*) shifted to the plasma membrane upon fusion to FRMD8 (*Figure 5E*). Furthermore, we found that the localisation of ADAM17-V5 followed that of iRhom in both conditions: in iRhom2$^{\Delta300}$ cells ADAM17 localised to the ER, and in FRMD8-iRhom2$^{\Delta300}$ cells it was readily observed at the cell surface. We also noted that FRMD8-iRhom2$^{\Delta300}$ showed strikingly higher total levels of iRhom2 (*Figure 5E*), which hinted that FRMD8 may play a role in the protein turnover of iRhoms at the cell surface. In line with these observations, we found that FRMD8-iRhom2$^{\Delta300}$ was much more stable compared to iRhom2$^{\Delta300}$ or iRhom2WT as seen in cells incubated with cycloheximide (CHX) to block the synthesis of new proteins (*Figure 5F*). Taken together, these data suggest that FRMD8 binding to iRhom2 stabilises the iRhom2 pool in the late secretory pathway and increases the cell surface localisation of the iRhom2/ADAM17 sheddase complex.

## FRMD8 recruitment protects iRhom2/ADAM17 from lysosomal degradation

Previous studies have shown that the cytoplasmic N-terminal region of iRhom2 is required to prevent lysosomal degradation of ADAM17 (*Grieve et al., 2017*). Therefore, we questioned whether the absence of FRMD8 recruitment to the iRhom2 N-terminus led to delivery of iRhom and ADAM17 to lysosomes. By immunofluorescence microscopy, iRhom2$^{WT}$ localisation is indistinguishable from iRhom2$^{\Delta300}$ (*Figure 6A,B*) within the endoplasmic reticulum (*Figure 1—figure supplement 1A*). However, upon treatment with the lysosomal degradation inhibitor, bafilomycin A1, both proteins accumulated in LAMP1-positive lysosomal puncta (*Figure 6C,D*). This suggests that there is a constant turnover of iRhom2 through the endo-lysosomal pathway, with iRhoms presumably cycling via the plasma membrane, before being degraded. Interestingly, unlike the partial colocalisation between LAMP1 and iRhom2$^{WT}$ (*Figure 6C*), iRhom2$^{\Delta300}$ overlapped completely with LAMP1 (indicated by the arrows in *Figure 6D*). This confirmed that in the absence of FRMD8 recruitment, iRhom2 is constitutively sent to lysosomes. Importantly, this lysosomal pool of iRhom2$^{\Delta300}$ also colocalised with ADAM17-V5 after bafilomycin treatment (highlighted with arrows in *Figure 6F*). All

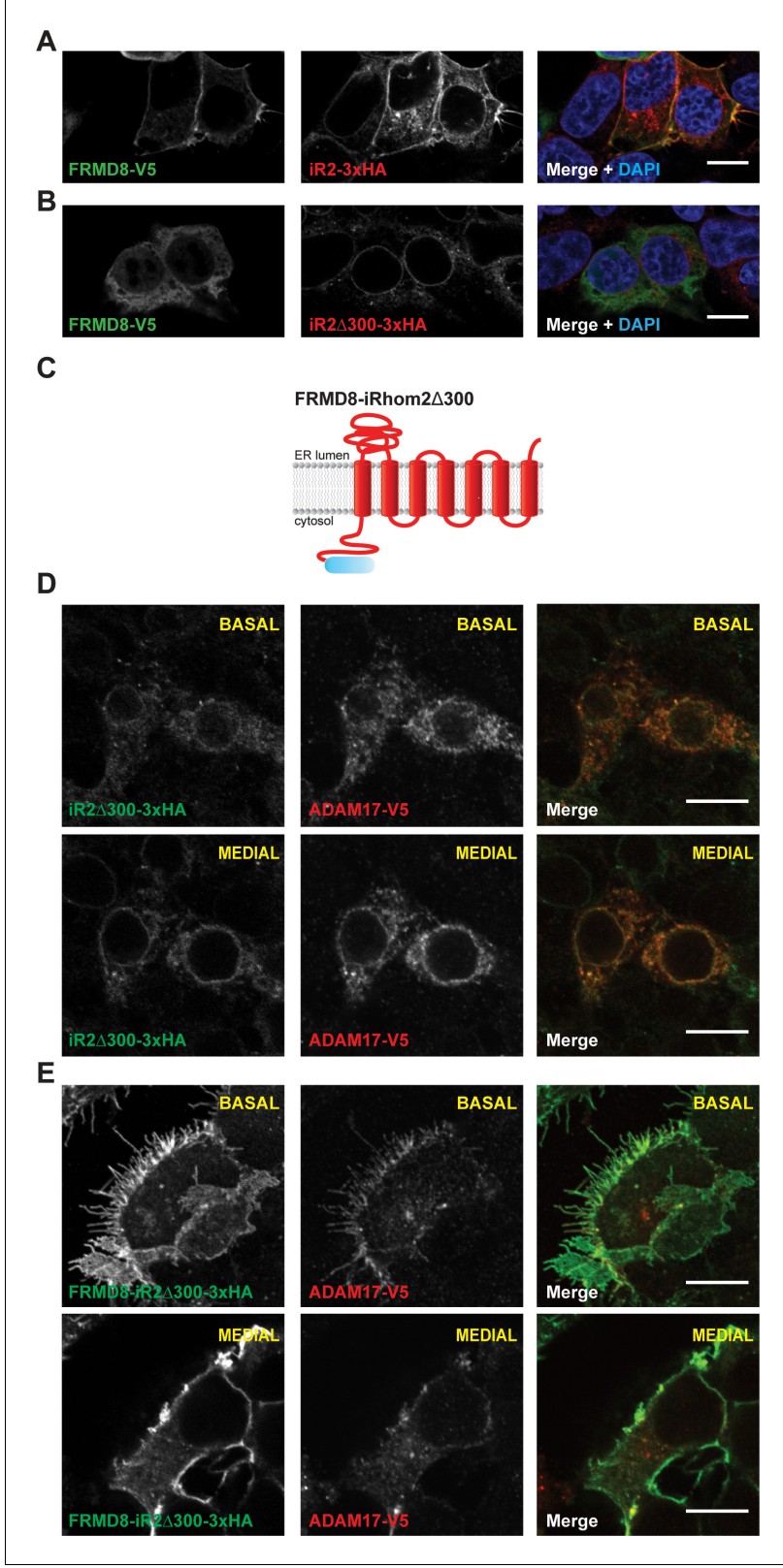

**Figure 5.** FRMD8 promotes cell surface localisation of iRhom2. (**A, B**) Immunofluorescence of iRhom1/2 double knockout HEK293T cells stably expressing iRhom2-3xHA or iRhom2$^{\Delta300}$-3xHA and transiently transfected with FRMD8-V5 for 72 hr. Cells were stained for HA (red), V5 (green) and DAPI for DNA (blue). Single confocal sections are shown, taken through the centre of the nucleus. (**C**) Schematic model of the FRMD8-iRhom2$^{\Delta300}$ construct used

*Figure 5 continued on next page*

*Figure 5 continued*

in (E). (D, E) Immunofluorescence of iRhom1/2 double knockout HEK293T cells stably expressing iRhom2$^{\Delta300}$-3xHA or FRMD8-iRhom2$^{\Delta300}$-3xHA and transiently transfected with ADAM17-V5 for 72 hr. Cells were stained for HA (green), V5 (red) and DAPI for DNA (blue). Single confocal sections are shown, taken either through the centre of the nucleus (MEDIAL), or at basal regions close to the coverslip (BASAL). In all images the scale bar = 10 μm.

DOI: https://doi.org/10.7554/eLife.35012.010

these data together indicate that the iRhom2/ADAM17 complex follow the same fate in the absence of FRMD8 recruitment (*Figure 6E,F*). Using a complementary approach, we tested the stability of ADAM17 in FRMD8 knockout cells. After 16 hr of treatment with the lysosomal degradation inhibitors bafilomycin and ammonium chloride, the mature form of ADAM17 was partially restored (*Figure 6G*; *Figure 7—figure supplement 1B*).

Combined, these results explain the reduced level of mature ADAM17 in FRMD8 knockout cells: it implies that the defect caused by loss of FRMD8 is not a failure of ADAM17 maturation, but instead a failure to stabilise the mature form. In line with this interpretation, the proteasomal inhibitor MG132 had no effect on the stability of mature ADAM17 (*Figure 7A*). We conclude that FRMD8 binding to iRhom2 acts to promote ADAM17 function by ensuring its stability after its maturation in the *trans*-Golgi network.

## FRMD8 functions to stabilise levels of iRhoms at the cell surface

If FRMD8 acts as a stabilising factor for the plasma membrane-localised iRhom2/ADAM17 sheddase complex, a difference in the cell surface level of iRhom2 is expected in the absence of FRMD8. Most tagged iRhom2 is ER-localised (*Figure 1—figure supplement 1A*, *Figure 6A*) and the cell surface fraction is relatively small (*Maney et al., 2015*; *Grieve et al., 2017*). Therefore, we used cell surface immunostaining of iRhom2 followed by flow cytometry to measure specifically the pool of iRhom2 at the cell surface. In the absence of FRMD8 we detected a significant loss of cell surface iRhom2 (*Figure 7A*). In line with our observation that cell surface iRhom2 represents only a small fraction of the total pool, a reduction of total iRhom2 levels was not detectable (*Figure 7B*). This further supports our observations that FRMD8 binding to iRhoms is required to stabilise the cell surface pool of iRhoms. Consistent with our conclusion that FRMD8 primarily functions late in the iRhom2/ADAM17 relationship, we detected no defects in the ER-based iRhom2/proADAM17 interaction in FRMD8 knockout cells (*Figure 4C*), nor in the trafficking of iRhom2 from the ER to the Golgi (*Figure 7—figure supplement 1C*).

Our results show that by binding to iRhom2, FRMD8 stabilises both iRhom2 and mature ADAM17, protecting them from degradation. A more direct demonstration of this stabilising function is provided by overexpressing FRMD8, which leads to increased levels of tagged iRhom2 (*Figure 7C*), as well as iRhom1 (*Figure 7—figure supplement 1D*). Note that the 50 kDa N-terminally truncated fragment of iRhoms detected in western blots (*Nakagawa et al., 2005*; *Adrain et al., 2012*; *Maney et al., 2015*) is not stabilised by FRMD8 expression (*Figure 7C*, *Figure 7—figure supplement 1D*). This iRhom fragment lacks the cytoplasmic tail, and therefore the binding site for FRMD8, so its insensitivity to FRMD8 is consistent with our model. Intriguingly, the stabilisation of iRhom2 and FRMD8 is mutual: overexpression of iRhom2 consistently led to the stabilisation of endogenous FRMD8 protein (*Figure 7D*) without affecting FRMD8 mRNA levels (*Figure 7E*). This indicates that the iRhom2-FRMD8 interaction leads to mutual stabilisation of both proteins as well as mutual effects on plasma membrane localisation (*Figure 5A*).

To ensure that our conclusion that FRMD8 stabilises iRhoms was not distorted by our use of overexpressed proteins, and in the absence of a usable antibody against human iRhom2, we used CRISPR/Cas9 to insert a triple HA tag into the *RHBDF2* locus to express endogenously C-terminally tagged iRhom2. siRNA-mediated knockdown of iRhom2 confirmed that this editing was successful (*Figure 8A*). The cells showed no defect in ADAM17 maturation (*Figure 8A*, *Figure 7—figure supplement 1E*), indicating that the tagged protein was functional. In these cells FRMD8 overexpression led to an increase in endogenous iRhom2 levels (*Figure 8A*); conversely, siRNA knockdown of FRMD8 caused a reduction of iRhom2 protein (*Figure 8B*), but no change of iRhom2 mRNA levels (*Figure 8C*). Again, the 50 kDa iRhom2 fragment was not affected by FRMD8 levels (*Figure 8A,B*).

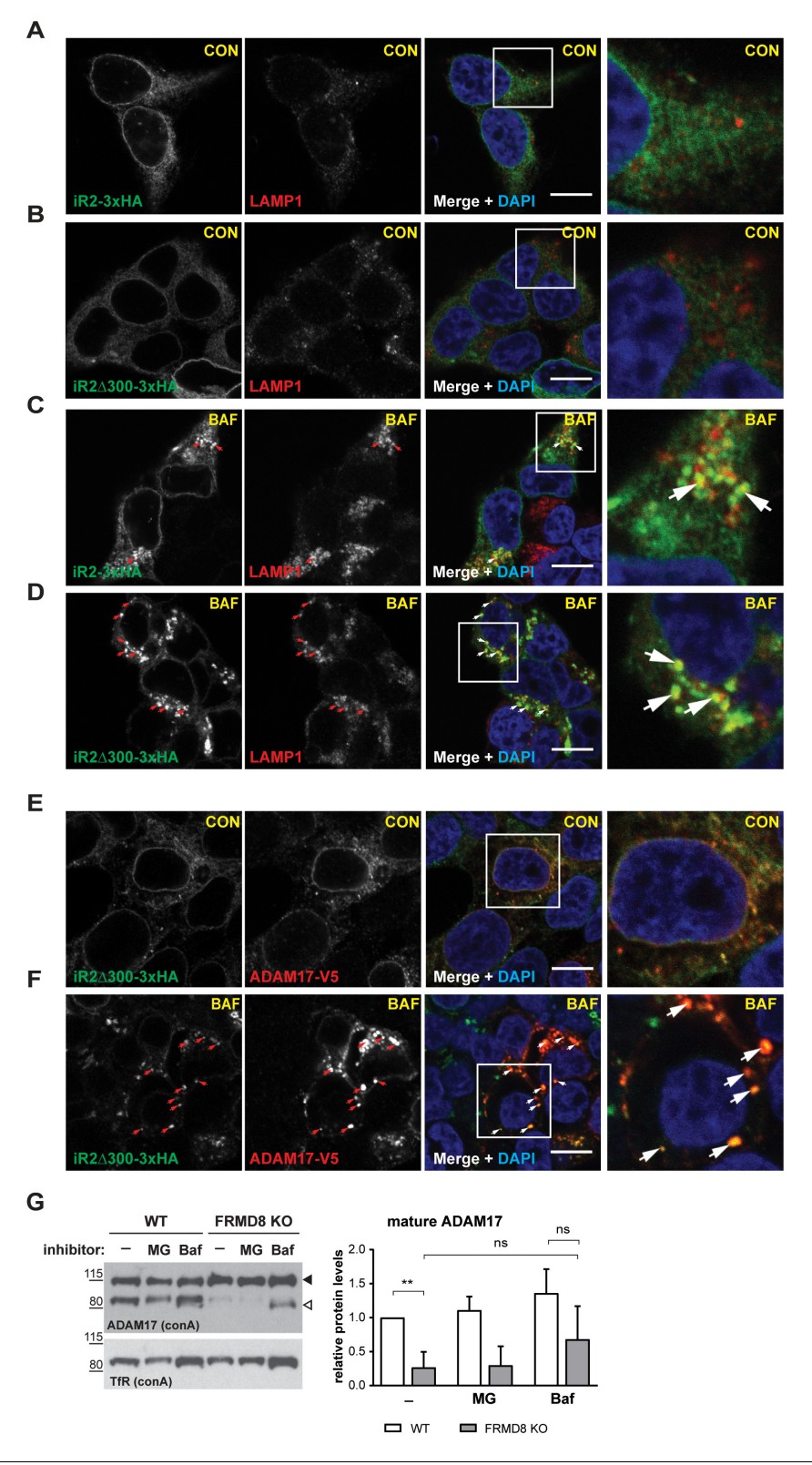

**Figure 6.** FRMD8 loss leads to degradation of iRhoms and mature ADAM17 through the lysosomal pathway. (A–D) Immunofluorescence of iRhom1/2 double knockout HEK293T cells stably expressing iRhom2-3xHA or iRhom2$^{\Delta300}$-3xHA treated with DMSO (CON) or 100 nM bafilomycin A1 (BAF) for 16 hr prior to fixation. Cells were stained for HA (green), the lysosomal marker LAMP1 (red) and DAPI for DNA (blue). LAMP1-labelled regions (within white boxes) have been magnified. Scale bar = 10 μm. (E, F) iRhom2$^{\Delta300}$-3xHA cells were treated as in (A–D), but with 72 hr expression of ADAM17-V5 and

*Figure 6 continued on next page*

*Figure 6 continued*

labelling of HA (green), V5 (red) and DAPI for DNA (blue). Arrows indicate colocalising puncta. Single confocal sections are shown, taken through the centre of the nucleus. HA- and V5-labelled regions (within white boxes) have been magnified. Scale bar = 10 µm. (G) Cell lysates of wild-type (WT) and FRMD8 knockout (KO) HEK293T cells treated with the solvent DMSO (–), 10 µM MG-132 (MG) or 200 nM bafilomycin A1 (Baf) for 16 hr were enriched for glycosylated proteins using concanavalin A (conA) beads and immunoblotted for ADAM17 and transferrin receptor 1 (TfR). TfR was used as a loading control although it is also susceptible to bafilomycin treatment. Mature ADAM17 levels from three experiments were quantified relative to TfR levels using ImageJ.

DOI: https://doi.org/10.7554/eLife.35012.011

Parenthetically, this is the first reported evidence that this iRhom fragment exists endogenously, although its functional significance remains unclear.

To summarise our results to this point, we have discovered that by binding to the iRhom2 cytoplasmic N-terminus, FRMD8 stabilises the cell surface iRhom2/ADAM17 sheddase complex. In the absence of FRMD8 recruitment to iRhom2, this enzyme complex is sent to lysosomes and degraded.

## FRMD8 binding to iRhom2 is essential for inflammatory signalling in human macrophages

We tested the pathophysiological significance of our conclusions by analysing the consequence of loss of FRMD8 in human macrophages, which release TNFα in response to tissue damage and inflammatory stimuli. To generate mutant human macrophages, we used CRISPR/Cas9 to knock out FRMD8 in an iPSC line that had previously been generated from dermal fibroblasts of a healthy female donor (*Fernandes et al., 2016*). The FRMD8 knockout and control iPSCs were analysed for deletions in the *FRMD8* gene by PCR (*Figure 9—figure supplement 1A*), and a normal karyotype was confirmed by single nucleotide polymorphism (SNP) analysis (*Figure 9—figure supplement 1B*) before differentiation into macrophages (*Figure 9A*). These mutant macrophages expressed no detectable FRMD8 and, as in the HEK293T cells, showed severely reduced levels of mature ADAM17 (*Figure 9B*). When challenged with the inflammatory trigger LPS, TNFα shedding from the cells, as measured by ELISA, was reduced (*Figure 9C*). Confirming the expected specificity, the ADAM10 inhibitor GI254023X (GI) had no effect on TNFα release from these cells, whereas GW, an inhibitor of both ADAM10 and ADAM17, further reduced TNFα release (*Figure 9—figure supplement 1C*). Although shedding was inhibited, TNFα expression by LPS was normal in these cells (*Figure 9—figure supplement 1D*). These results demonstrate that our conclusions about the requirement for FRMD8 in ADAM17 function in cell culture models does indeed apply to human macrophages.

## Loss of FRMD8 in mice highlights its physiological role in stabilising the iRhom/ADAM17 complex

To investigate further the physiological significance of our discovery of the role of FRMD8 in stabilising iRhom/ADAM17 sheddase complexes, we analysed the levels of ADAM17 and iRhom2 in tissues from FRMD8-deficient mice. These mice were generated from embryonic stem (ES) cells from the KOMP Repository, University of California Davis, in which all coding exons (2-11) of the *Frmd8* gene were deleted (*Figure 9—figure supplement 2A*). *Frmd8⁻ᐟ⁻* mice are viable (*Figure 9—figure supplement 2B*) and fertile. The knockout was confirmed by western blot (*Figure 9D*). Western blot analysis of tissues of *Frmd8⁻ᐟ⁻* mice showed that mature ADAM17 levels were reduced in all tissues examined compared to tissues from wild-type littermates (*Figure 9D*). This confirms that FRMD8 controls the level of mature ADAM17 *in vivo*. Of note, there was a major reduction of mature ADAM17 levels in the brain, a tissue in which iRhom2 in almost completely absent but iRhom1 levels are high (*Christova et al., 2013*; *Li et al., 2015*). This supports our hypothesis that FRMD8 regulates mature ADAM17 levels through iRhom1 as well as iRhom2. We also tested *in vivo* our conclusion that FRMD8 loss destabilises endogenous iRhoms (*Figure 8B*). Using an antibody that we had previously generated against mouse iRhom2 (*Adrain et al., 2012*), we analysed iRhom2 levels in *Frmd8⁺ᐟ⁺* and *Frmd8⁻ᐟ⁻* mouse tissues. In lung and skin, both tissues with high iRhom2 expression (*Christova et al., 2013*), we detected a strong decrease of iRhom2 protein levels in *Frmd8⁻ᐟ⁻* compared to wild-type (*Figure 9E*, *Figure 9—figure supplement 2C*). Tissue from *Rhbdf2⁻ᐟ⁻* mice served as a control for the iRhom2 antibody specificity (*Figure 9E*, *Figure 9—figure supplement 2C*). The

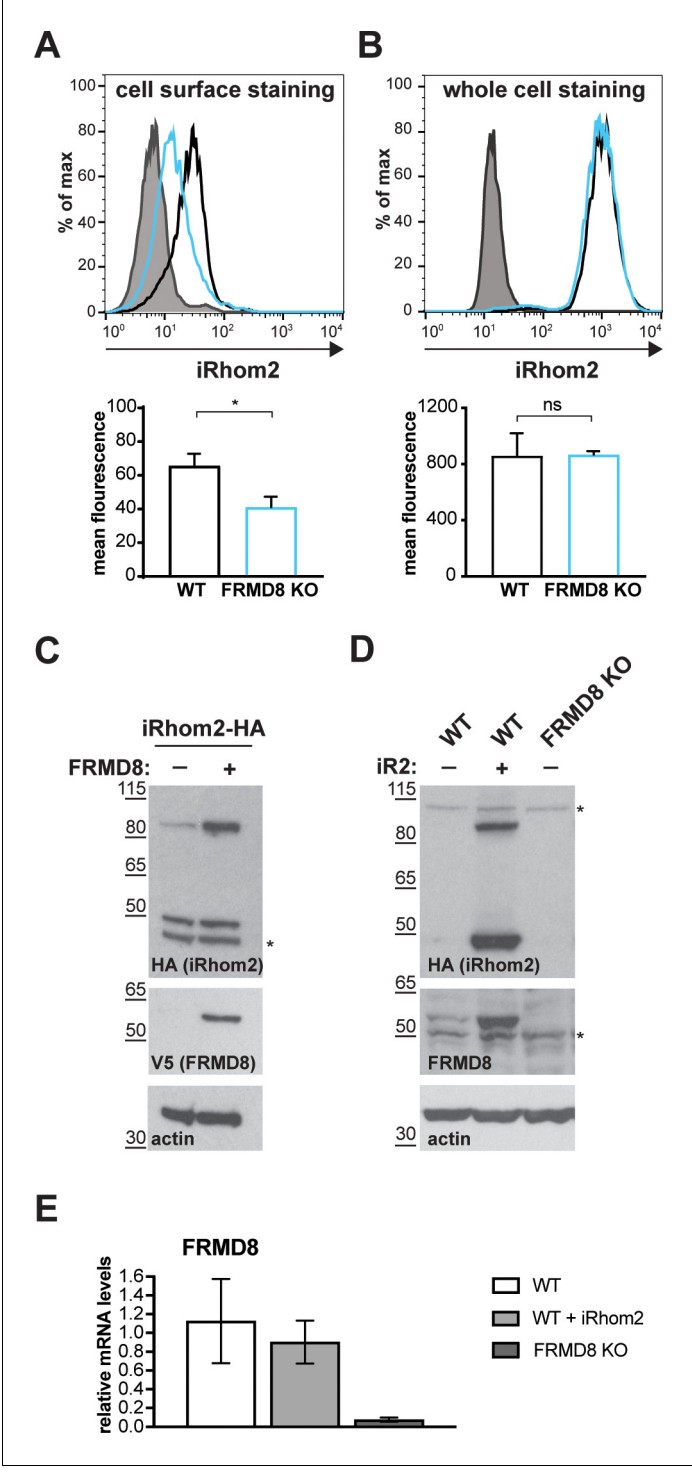

**Figure 7.** FRMD8 loss leads to the destabilisation of ADAM17 and iRhom2. (**A**) Unpermeabilised WT (black) and FRMD8 KO HEK293T (cyan) cells stably expressing human iRhom2-3xHA were immunostained on ice for HA. Wild-type HEK293T cells immunostained for HA served as a negative control (grey). (**B**) Cells were permeabilised and stained at room temperature with an anti-HA antibody. Immunostaining with the Alexa Fluor 488-coupled secondary antibody served as a control (grey). The flow cytometry graphs shown are one representative experiment out of three experiments. The geometric mean fluorescence was calculated for each experiment using FlowJo software. Statistical analysis was performed using an unpaired t-test; ns = p value>0.05; *=p value<0.05. (**C**) Lysates of HEK293T cells stably expressing human iRhom2-3xHA and transiently transfected with FRMD8-V5 (where indicated) were analysed by western blot for iRhom2 levels using anti-HA, anti-V5 and anti-actin

*Figure 7 continued on next page*

*Figure 7 continued*
immunostaining. Nonspecific bands are marked with an asterisk. (D) Lysates of WT and FRMD8 KO HEK293T cells
stably expressing human iRhom2-3xHA (where indicated) were immunoblotted for HA, FRMD8 and actin. An
asterisk marks nonspecific bands. (E) FRMD8 mRNA levels relative to actin mRNA levels were determined by
TaqMan PCR in cells used in (D).
DOI: https://doi.org/10.7554/eLife.35012.012
The following figure supplement is available for figure 7:

**Figure supplement 1.** FRMD8 stabilises iRhom levels by preventing its lysosomal degradation.
DOI: https://doi.org/10.7554/eLife.35012.013

reduction of endogenous iRhom2 and mature ADAM17 levels in mouse lung was about 75%
(*Figure 9E*), which is comparable to the reduction of mature ADAM17 levels in iPSC-derived human
macrophages (*Figure 9B*). In summary, our experiments in mice confirm the physiological impor-
tance of our prior conclusions: FRMD8 is required *in vivo* to regulate the stability of the iRhom/
ADAM17 sheddase complex and is therefore a previously unrecognised essential component in reg-
ulating cytokine and growth factor signalling.

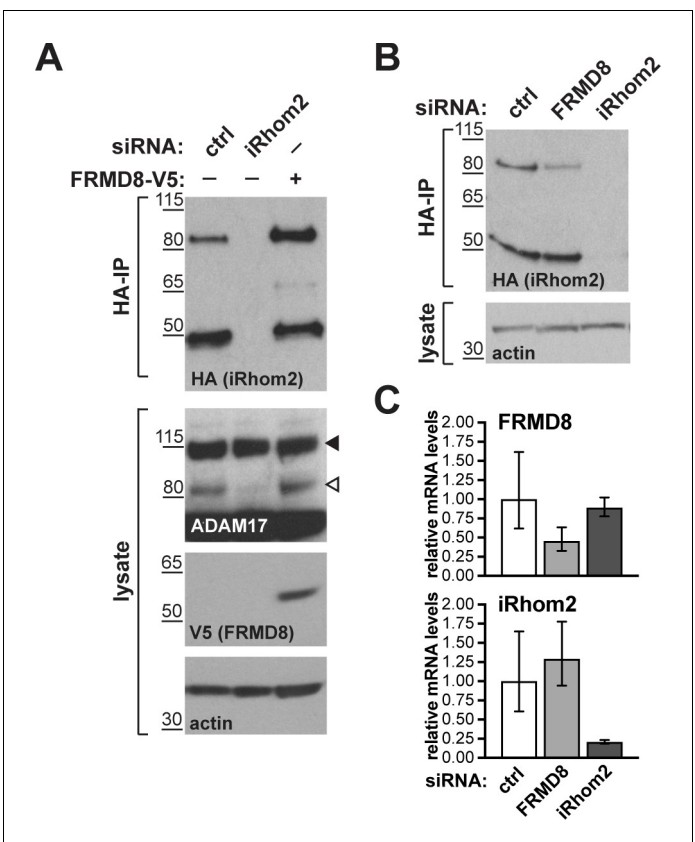

**Figure 8.** FRMD8 stabilises endogenous iRhom2. (A, B) Levels of endogenously 3xHA tagged iRhom2 were
analysed in HEK293T-iRhom2-3xHA cells transfected with FRMD8-V5 plasmid, siRNAs targeting iRhom2, non-
targeting siRNA control pool (ctrl) or FRMD8 SMARTpool siRNA. Cell lysates were anti-HA immunoprecipitated
(HA-IP) to detect endogenous iRhom2-3xHA levels and immunoblotted using anti-HA antibody. Cell lysates were
immunoblotted for ADAM17, V5, and actin. (C) FRMD8 and iRhom2 mRNA levels relative to actin mRNA levels
were determined by TaqMan PCR in cells used for the experiment shown in (B) to demonstrate that the
destabilisation of endogenous iRhom2 was not induced by a change in iRhom2 mRNA levels.
DOI: https://doi.org/10.7554/eLife.35012.014

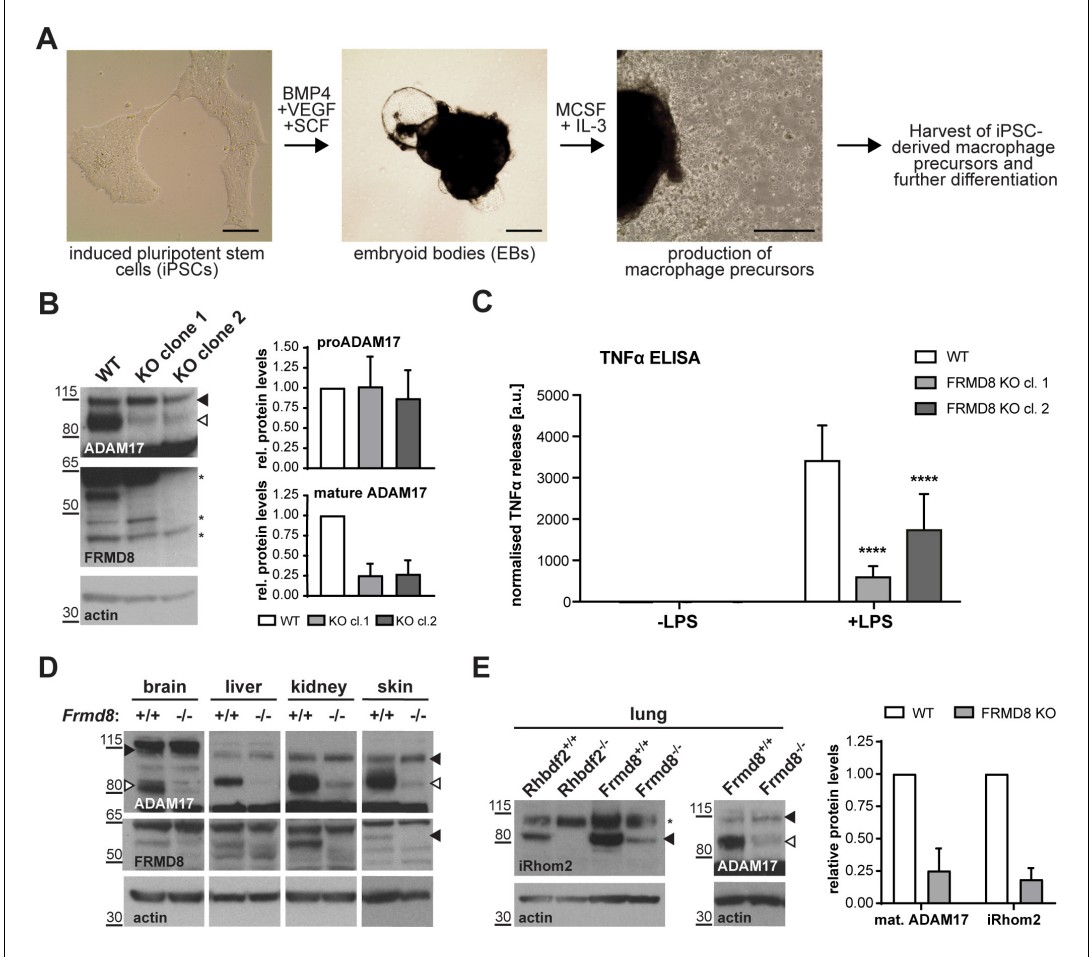

**Figure 9.** FRMD8 is required for iRhom2/TACE regulation in human iPSC-derived macrophages and mice . (**A**) Schematic representation of the differentiation protocol of iPSCs into macrophages based on (**van Wilgenburg et al., 2013**). Scale bars = 10 µm. (**B**) Lysates of iPSC-derived macrophages (on day seven after harvest from EBs) were immunoblotted for ADAM17, FRMD8, and actin. Western blots from three experiments were quantified using ImageJ with actin serving as the loading control. (**C**) 25,000 iPSC-derived macrophages were either left unstimulated or stimulated with 50 ng/ml LPS for 4 hr. TNFα concentration in the cell supernatants was measured by ELISA and then normalised to the protein concentration in macrophage cell lysates to adjust the cytokine release for potential differences in cell numbers. Each experiment was performed in biological triplicates. Data from three independent experiments were statistically analysed using a Mann-Whitney test; ***=p value<0.001; ****=p value<0.0001. (**D, E**) Lysates from tissues derived from *Frmd8$^{-/-}$* or *Rhbdf2$^{-/-}$* and their wild-type littermates were immunoblotted for ADAM17, FRMD8, iRhom2 and actin. Blots from three experiments using three different littermates of *Frmd8$^{-/-}$* and *Frmd8$^{+/+}$* mice were quantified using ImageJ with actin serving as the loading control.

DOI: https://doi.org/10.7554/eLife.35012.015

The following figure supplements are available for figure 9:

**Figure supplement 1.** Generation of FRMD8 knockout iPSCs and iPSC-derived macrophages.

DOI: https://doi.org/10.7554/eLife.35012.016

**Figure supplement 2.** Generation of Frmd8 knockout mice.

DOI: https://doi.org/10.7554/eLife.35012.017

# Discussion

ADAM17 is the shedding enzyme that is responsible for not only the activation of inflammatory TNFα signalling, but also the release from the cell surface of multiple EGF family growth factors and other proteins. Its regulation has therefore received much attention, both from the perspective of fundamental cell biology and because of the proven therapeutic significance of blocking TNFα (**Monaco et al., 2015**). Here we report that FRMD8 is a new component of the regulatory machinery that controls the release of ADAM17 substrates, including TNFα. We identified FRMD8 as a

prominent binding partner of iRhoms, which are rhomboid-like proteins that act as regulatory cofactors of ADAM17. Our subsequent experiments demonstrate that although FRMD8 binds to iRhoms throughout their life cycle, its function appears to be confined to the later stages of their role in regulating ADAM17. FRMD8 stabilises the iRhom2/ADAM17 complex at the cell surface, ensuring it is available to shed TNFα and growth factors. We took advantage of iPSC technology to generate human FRMD8 knockout macrophages, allowing us to confirm that the mechanistic conclusions derived mostly from HEK293T cell models were indeed relevant to the human cells that provide the primary inflammatory response. Finally, tissues from FRMD8 knockout mice demonstrate the physiological importance of FRMD8 in a whole organism, and confirm that it stabilises the iRhom/mature ADAM17 complex *in vivo*.

Bringing together all our results, we propose the following model of FRMD8 function in ADAM17-dependent signalling: FRMD8 binds to the cytoplasmic domain of iRhoms throughout the secretory pathway, forming a tripartite complex when iRhoms are also bound to ADAM17. Despite this long-term relationship, we have found no evidence for a functional role for FRMD8 in ER-to-Golgi trafficking or ADAM17 maturation. Instead, FRMD8 acts later, to prevent the endolysosomal degradation of the iRhom/ADAM17 complex (*Figure 10*). The exact molecular detail of FRMD8 action on the iRhom2/ADAM17 sheddase complex is unclear. It is possible that FRMD8 increases the delivery of the iRhom2/ADAM17 sheddase complex from the Golgi apparatus to the cell surface, stabilises the complex by preventing its internalisation, or promotes the endosomal retrieval to the cell surface. In all cases, it is likely that the recruitment of additional proteins is required. Therefore, understanding the molecular interactions of FRMD8, as well as the FRMD8/iRhom2/mature ADAM17 complex at the cell surface, will shed light into the molecular mechanism.

As we have previously reported, it is the iRhom2/ADAM17 complex that is responsible for shedding ADAM17 substrates including TNFα. Without FRMD8, iRhoms and mature ADAM17 are destabilised and the cell cannot shed TNFα in response to an inflammatory challenge. Combined with our

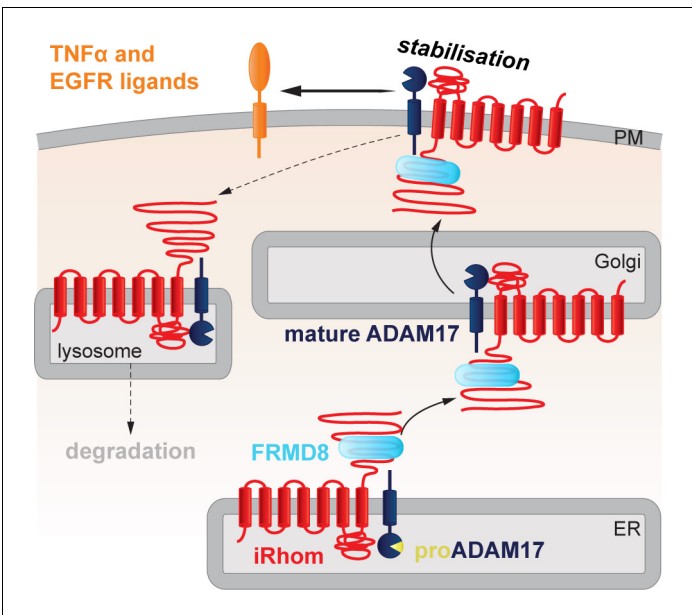

**Figure 10.** FRMD8 stabilises the iRhom2/ADAM17 sheddase complex at the cell surface. Schematic representation of the role of FRMD8 in the iRhom2/ADAM17 pathway: under wild-type conditions ADAM17 and iRhom2 are stabilised by FRMD8 and thereby protected from degradation through the endolysosmal pathway.
DOI: https://doi.org/10.7554/eLife.35012.018

The following figure supplement is available for figure 10:

**Figure supplement 1.** Lysates and anti-HA immunoprecipitation (HA-IP) from HEK293T cells transiently transfected with human FRMD8-V5 and mouse iRhom2^WT (WT) or iRhom2^pDEAD (pDEAD) were immunoblotted for V5 and HA.
DOI: https://doi.org/10.7554/eLife.35012.019

previous studies (*Grieve et al., 2017*), this work has changed our perspective on ADAM17, the central enzyme in cytokine and growth factor shedding. Our evidence implies that it would be more appropriate to consider it as the active subunit of a regulatory complex at the cell surface, where iRhoms provide regulatory functions (*Maney et al., 2015*; *Cavadas et al., 2017*; *Grieve et al., 2017*), and FRMD8 maintains the stability of the iRhom/ADAM17 complex post-ADAM17 maturation. It is essential that a pool of the sheddase is available on the cell surface to execute, for example, rapid cytokine release in response to inflammatory signals induced by bacterial infection.

In the only other paper about FRMD8 function, it was reported that FRMD8 (named Bili, after the Drosophila mutation) negatively regulates Wnt signalling by binding to the LRP6 co-receptor, thereby preventing the recruitment of the signal transduction protein axin (*Kategaya et al., 2009*). Although the signalling event being regulated is different, there is the obvious parallel that in both cases FRMD8 binds to the cytoplasmic tail of a transmembrane protein. In the case of Wnt signalling, this prevents the recruitment of axin; in the case of iRhom function, we do not yet know what the next step in the molecular chain of events is, but the cellular consequence is to prevent recruitment of iRhoms into the endolysosomal degradation system.

Our results extend an important theme to emerge from a number of studies, namely the significance of the iRhom cytoplasmic N-terminal region in regulating iRhom/ADAM17 function. Several reports indicate that N-terminal mutations in iRhoms cause complex phenotypes that combine aspects of gain and loss of iRhom function, which is consistent with a regulatory function for this region. First, the *cub* mutation, an N-terminal deletion in mouse iRhom2, does not abolish protein function but instead modulates it in complex ways that are still poorly understood (*Hosur et al., 2014*; *Siggs et al., 2014*). *cub* was described as a gain-of-function mutation that leads to constitutively elevated release of amphiregulin, but is also reported to be defective in releasing TNFα (*Hosur et al., 2014*). Second, specific point mutations in the N-terminus of human iRhom2 are the cause of a rare genetic disorder called tylosis with oesophageal cancer (TOC) (*Blaydon et al., 2012*; *Saarinen et al., 2012*). TOC mutations, as well as truncation of parts of the N-terminus have been reported to enhance the activity of ADAM17 (*Maney et al., 2015*), leading to the conclusion that parts of the N-terminus have inhibitory functions on ADAM17 function. Third, phosphorylation of specific sites in the iRhom2 N-terminus result in 14-3-3 binding and consequent activation of substrate shedding by associated ADAM17 (*Grieve et al., 2017*; *Cavadas et al., 2017*), demonstrating that the N-terminus of iRhom2 also positively regulates ADAM17. The FRMD8 binding region does not overlap with these sites required for phosphorylation-dependent 14-3-3 binding, however it is formally possible that there is some functional overlap between them. We could not detect major changes in the interaction of FRMD8 with iRhom2 upon PMA stimulation (*Figure 10—figure supplement 1*), which leads to the phosphorylation of iRhoms (*Grieve et al., 2017*; *Cavadas et al., 2017*). Moreover, an iRhom2 mutant, in which 15 conserved phosphorylation sites have been mutated to alanine (iRhom2$^{pDEAD}$; *Figure 3—figure supplement 1A*) (*Grieve et al., 2017*), did not abolish the interaction with FRMD8 (*Figure 10—figure supplement 1*). This conclusively demonstrates that the binding of FRMD8 to iRhom2 does not require phosphorylation of iRhom2. However, it is still formally possible that phosphorylation of iRhom2 affects FRMD8 binding specifically at the cell surface. This change cannot be detected by analysing the entire iRhom2 pool, which is primarily localised in the early secretory pathway. Therefore, we cannot exclude that the phosphorylation state of the relatively small cell surface pool of iRhom2 regulates the interaction with FRMD8.

Consistent with our current results, we reported previously that iRhom2 lacking the entire N-terminus is not sufficient to support ADAM17-mediated shedding in iRhom1/2-deficient cells, although it can promote ER-to-Golgi trafficking of ADAM17 (*Grieve et al., 2017*). Complementary to the conclusion that iRhom N-termini are regulatory, the core TMD binding function of iRhoms depends on their membrane-embedded region (*Grieve et al., 2017*; *Cavadas et al., 2017*). A picture therefore begins to emerge of iRhoms having a modular structure, with a core, highly conserved TMD recognition domain in the membrane (and perhaps the lumen), regulated by a more variable N-terminal domain that can integrate cytoplasmic signals.

In light of the growing value of therapeutics that block TNFα signalling, and the wider potential of modulating a wide range of ADAM17 substrates, it is tempting to speculate that the cytoplasmic N-termini of iRhoms could provide potential new drug target opportunities. For example, the limited expression of iRhom2 makes it a theoretically attractive anti-inflammatory target (*Issuree et al., 2013*; *Lichtenthaler, 2013*). iRhom2 knockout mice are broadly healthy, beyond defects in TNFα

and type I interferon signalling that are only apparent upon challenge by bacterial and viral infections (*McIlwain et al., 2012*; *Luo et al., 2016*). Our work now implies that the interface between FRMD8 and iRhoms might be a useful target. This is supported, at least in principle, by our observation that even in cells with complete loss of FRMD8, there is still a low level of mature ADAM17 at the cell surface, and consequently residual TNFα shedding. Even very efficient pharmacological blocking of the FRMD8/iRhom interaction would not, therefore, fully abolish inflammatory responses, potentially reducing side effects. Consistent with this idea, mice with a hypomorphic mutation in ADAM17 (termed ADAM17$^{ex/ex}$) show that even only 5% of normal ADAM17 expression is sufficient to rescue many aspects of the loss of function phenotype (*Chalaris et al., 2010*). Moreover, a recent study has shown that reducing ADAM17 levels has great pharmaceutical potential: reduced levels of ADAM17 in the *Adam17$^{ex/ex}$* mouse limits colorectal cancer formation and any residual tumours are low-grade dysplasias (*Schmidt et al., 2018*).

In conclusion, our work demonstrates the cellular and physiological significance of FRMD8 binding to iRhoms, and how it stabilises the iRhom/ADAM17 sheddase complex at the cell surface. It also reinforces the picture that has begun to emerge of ADAM17 not acting alone but instead being supported by at least two other regulatory proteins that act as subunits of what is effectively an enzyme complex. This concept would help to explain how the activity of such a powerful and versatile – and therefore potentially dangerous – shedding enzyme is controlled with necessary precision. The next steps in fully revealing the role of FRMD8 will be to analyse the phenotypic consequences of its loss in mice, which should allow us to understand how the roles of FRMD8 in ADAM17 activation, Wnt signalling, and any other potential functions, are integrated. Notwithstanding these physiological questions, the work described here already provides a basis for beginning to investigate the potential of targeting the FRMD8/iRhom interface for modulating the release of ADAM17 substrates.

# Materials and methods

**Key resources table**

| Reagent type (species) or resource | Designation | Source or reference | Identifiers | Additional information |
|---|---|---|---|---|
| Strain, strain background (mouse) | Frmd8$^{-/-}$: C57BL/6-Frmd8$^{tm1(KOMP)Vlcg}$ | This paper | N/A | FRMD8 KO mice generated as described in materials and methods |
| Strain, strain background (mouse) | WT control: C57BL/6 | This paper | N/A | WT control for FRMD8 KO mice |
| Strain, strain background (mouse) | Rhbdf2$^{-/-}$: C57BL/6 -Rhbdl6$^{A22}$ | (*Adrain et al., 2012*) | N/A | |
| Cell line (human) | HEK293T cells | Freeman lab | RRID:CVCL_0063 | |
| Cell line (human) | HEK293T human iRhom1$^{WT}$ | This paper | N/A | HEK293T cells transduced with pLEX.puro-human iRhom1WT-3xHA |
| Cell line (human) | HEK293T human iRhom2$^{WT}$ | This paper | N/A | HEK293T cells transduced with pLEX.puro-human iRhom2WT-3xHA |
| Cell line (human) | HEK293T human UNC93B1 | This paper | N/A | HEK293T cells transduced with pLEX.puro-human UNC93B1-3xHA |
| Cell line (human) | HEK293T FRMD8 KO | This paper | N/A | CRISPR/Cas9-mediated KO cell line as described in materials and methods |
| Cell line (human) | HEK293T FRMD8 KO + human iRhom2$^{WT}$ | This paper | N/A | FRMD8 KO cells transduced with pLEX.puro-human iRhom2WT-3xHA |
| Cell line (human) | HEK293T endogenous iRhom2-3xHA | This paper | N/A | CRISPR/Cas9-mediated knock-in cell line as described in materials and methods |

*Continued on next page*

*Continued*

| Reagent type (species) or resource | Designation | Source or reference | Identifiers | Additional information |
|---|---|---|---|---|
| Cell line (human) | HEK293T iRhom1/iRhom2 double-knockout (DKO) | This paper | N/A | CRISPR/Cas9-mediated KO cell line as described in materials and methods |
| Cell line (human) | HEK293T iRhom1/iRhom2 DKO + human iRhom2WT | This paper | N/A | DKO cells transduced with pLEX.puro-human iRhom2WT-3xHA |
| Cell line (human) | HEK293T iRhom1/iRhom2 DKO + human iRhom2Δ300 | This paper | N/A | DKO cells transduced with pLEX.puro-human iRhom2Δ300-3xHA |
| Cell line (human) | HEK293T iRhom1/iRhom2 DKO + human FRMD8-iRhom2Δ300 | This paper | N/A | DKO cells transduced with pLEX.puro-human FRMD8-iRhom2Δ300-3xHA |
| Cell line (human) | HEK293T iRhom1/iRhom2 DKO + human iRhom2Δ 201–300 | This paper | N/A | DKO cells transduced with pLEX.puro-human iRhom2Δ201-300-3xHA |
| Cell line (human) | hiPSC line AH017-13 | (*Fernandes et al., 2016*) | James Martin Stem Cell Facility | |
| Cell line (human) | hiPSC line AH017-13 FRMD8 KO clone 1 (clone F1) | This paper | N/A | CRISPR/Cas9-mediated KO cell line as described in materials and methods |
| Cell line (human) | hiPSC line AH017-13 FRMD8 KO clone 2 (clone G6) | This paper | N/A | CRISPR/Cas9-mediated KO cell line as described in materials and methods |
| Cell line (human) | hiPSC line AH017-13 WT (clone E4) | This paper | N/A | unedited WT control for FRMD8 KO iPSCs clones |
| Cell line (mouse) | Frmd8$^{-/-}$ ES cells: C57BL/6NTac-Frmd8$^{tm1(KOMP)Vlcg}$ (clone 17364AC3) | KOMP | RRID:IMSR_KOMP: VG17364-1-Vlcg | |
| Antibody | anti-β-actin-HRP, mouse monoclonal (clone AC-15) | Sigma-Aldrich | Cat#A3854; RRID:AB_262011 | dilution is described in materials and methods |
| Antibody | anti-ADAM10, mouse monoclonal (clone SHM14) | BioLegend | Cat#352702; RRID:AB_10897813 | dilution is described in materials and methods |
| Antibody | anti-ADAM10, rabbit polyclonal | Cell Signaling Technology | Cat#14194 | dilution is described in materials and methods |
| Antibody | anti-ADAM17, mouse monoclonal (clone A300E) | (*Yamamoto et al., 2012*); received from Stefan Düsterhöft | N/A | dilution is described in materials and methods |
| Antibody | anti-ADAM17, rabbit polyclonal | Abcam | Cat#ab39162; RRID:AB_722565 | dilution is described in materials and methods |
| Antibody | anti-calnexin, rabbit polyclonal | Santa Cruz | Cat#sc-11397; RRID:AB_2243890 | dilution is described in materials and methods |
| Antibody | anti-FRMD8, rabbit polyclonal | Abcam | Cat#ab169933 | dilution is described in materials and methods |
| Antibody | anti-HA, rabbit polyclonal | Santa Cruz | Cat#sc-805; RRID:AB_631618 | dilution is described in materials and methods |
| Antibody | anti-HA, rat monoclonal (clone 3F10) | Roche | Cat#11867423001; RRID:AB_10094468 | dilution is described in materials and methods |
| Antibody | anti-HA tag, rabbit monoclonal (C29F4) | CST | Cat#3724; RRID:AB_1549585 | dilution is described in materials and methods, used at 1:1000 for IF |
| Antibody | anti-HA-HRP, rat monoclonal (clone 3F10) | Roche | Cat#12013819001; RRID:AB_390917 | dilution is described in materials and methods |
| Antibody | anti-iRhom2, rabbit polyclonal | (*Adrain et al., 2012*) | N/A | dilution is described in materials and methods |

*Continued on next page*

*Continued*

| Reagent type (species) or resource | Designation | Source or reference | Identifiers | Additional information |
|---|---|---|---|---|
| Antibody | ant-LAMP1, mouse monoclonal (H4A3) | Santa Cruz | Cat#sc-20011; RRID:AB_626853 | dilution is described in materials and methods, used at 1:250 for IF |
| Antibody | anti-transferrin receptor 1, mouse monoclonal (clone H68.4) | Thermos Fisher Scientific | Cat#13–6800; RRID:AB_86623 | dilution is described in materials and methods |
| Antibody | anti-V5, goat polyclonal | Santa Cruz | Cat#sc-83849; RRID:AB_2019670 | dilution is described in materials and methods, used at 1:1000 for IF |
| Antibody | anti-goat-HRP, mouse monoclonal | Santa Cruz | Cat#sc-2354; RRID:AB_628490 | dilution is described in materials and methods |
| Antibody | anti-mouse-HRP, goat polyclonal | Santa Cruz | Cat#sc-2055; RRID:AB_631738 | dilution is described in materials and methods |
| Antibody | anti-rabbit-HRP, goat polyclonal | Sigma-Aldrich | Cat#A9169; RRID:AB_258434 | dilution is described in materials and methods |
| Antibody | anti-mouse Alexa Fluor 488, donkey polyclonal | Thermos Fisher Scientific | Cat#A-21202; RRID:AB_141607 | dilution is described in materials and methods |
| Antibody | anti-rabbit Alexa Fluor 488, donkey polyclonal | Thermos Fisher Scientific | Cat#A-21206; RRID:AB_2535792 | dilution is described in materials and methods |
| Antibody | anti-rabbit Alexa Fluor 647, donkey polyclonal | Thermos Fisher Scientific | Cat#A-31573; RRID:AB_2536183 | dilution is described in materials and methods |
| Recombinant DNA reagent | cDNA: human iRhom2 (NM_024599.2) | Origene | Cat#SC122961 | |
| Recombinant DNA reagent | cDNA: human FRMD8 (NM_031904) | Origene | Cat#SC107202 | |
| Recombinant DNA reagent | cDNA: human UNC93B1 | (*Brinkmann et al., 2007*) | N/A | |
| Recombinant DNA reagent | Plasmid: pLEX.puro | Thermo Fisher Scientific | Cat#OHS4735 | |
| Recombinant DNA reagent | Plasmid: pcDNA3.1(+) | Thermo Fisher Scientific | Cat#V790-20 | |
| Recombinant DNA reagent | Plasmid: entiviral packaging plasmid | (*Adrain et al., 2012*) | N/A | |
| Recombinant DNA reagent | Plasmid: entiviral envelope plasmid | (*Adrain et al., 2012*) | N/A | |
| Recombinant DNA reagent | Plasmid: pSpCas9(BB)—2A-Puro (pX459) | (*Ran et al., 2013*) | Addgene plasmid #48139 | |
| Recombinant DNA reagent | Plasmid: epX459(1.1) | received from Joey Riepsaame | N/A | |
| Recombinant DNA reagent | Plasmid: pSpCas9(BB)—2A-Puro V2.0 (pX462 V2.0) | (*Ran et al., 2013*) | Addgene plasmid #62987 | |
| Transfected construct (human) | Plasmid: pLEX.puro-human iRhom1WT-3xHA | (*Christova et al., 2013*) | N/A | |
| Transfected construct (human) | Plasmid: pLEX.puro-human iRhom2WT-3xHA | This paper | N/A | cloned from human iRhom2 (NM_024599.2) |
| Transfected construct (human) | Plasmid: pLEX.puro-human iRhom2Δ100-3xHA | This paper | N/A | human iRhom2 lacking amino acids 1–100 |
| Transfected construct (human) | Plasmid: pLEX.puro-human iRhom2Δ200-3xHA | This paper | N/A | human iRhom2 lacking amino acids 1–200 |
| Transfected construct (human) | Plasmid: pLEX.puro-human iRhom2Δ300-3xHA | This paper | N/A | human iRhom2 lacking amino acids 1–300 |
| Transfected construct (human) | Plasmid: pLEX.puro-human iRhom2Δ201-300-3xHA | This paper | N/A | human iRhom2 lacking amino acids 1–382 |

*Continued on next page*

*Continued*

| Reagent type (species) or resource | Designation | Source or reference | Identifiers | Additional information |
|---|---|---|---|---|
| Transfected construct (human) | Plasmid: pLEX.puro-human iRhom2Δ382-3xHA | This paper | N/A | human iRhom2 lacking amino acids 201–300 |
| Transfected construct (human) | Plasmid: pLEX.puro-human FRMD8-iRhom2Δ300-3xHA | This paper | N/A | human FRMD8 fused to human iRhom2Δ300 via a flexible linker (GSGSGS) |
| Transfected construct (mouse) | Plasmid: pM6P.blast-mouse iRhom2$^{WT}$-3xHA | (*Grieve et al., 2017*) | N/A | |
| Transfected construct (mouse) | Plasmid: pM6P.blast-mouse iRhom2$^{cub}$-3xHA | (*Grieve et al., 2017*) | N/A | |
| Transfected construct (mouse) | Plasmid: pM6P.blast-mouse iRhom2$^{pDEAD}$-3xHA | (*Grieve et al., 2017*) | N/A | |
| Sequence-based reagent | gRNA targeting exon 7 of human FRMD8 (ACCC ATAAAACGGCAGCTCG) | This paper | N/A | gRNA targeting exon 7 of human FRMD8 |
| Sequence-based reagent | gRNA targeting exon 19 of human RHBDF2 (AG CGGTCAGTGCAGCACCT) | This paper | N/A | gRNA targeting exon 19 of human RHBDF2 |
| Sequence-based reagent | gRNA targeting exon 3 of human RHBDF1 (GGAACC ATGAGTGAGGCCCC) | This paper | N/A | gRNA targeting exon 3 of human RHBDF1 |
| Sequence-based reagent | gRNA targeting exon 3 of human RHBDF1 (GGGTGG CTTCTTGCGCTGCC) | This paper | N/A | gRNA targeting exon 3 of human RHBDF1 |
| Sequence-based reagent | gRNA targeting exon 10 of human RHBDF1 (AGCCGT GTGCATCTATGGCC) | This paper | N/A | gRNA targeting exon 10 of human RHBDF1 |
| Sequence-based reagent | gRNA targeting exon 10 of human RHBDF1 (CCGTCTC ATGCTGCGAGAAC) | This paper | N/A | gRNA targeting exon 10 of human RHBDF1 |
| Sequence-based reagent | gRNA targeting exon 2 of human RHBDF2 (GCAGAG CCGGAAGCCACCCC) | This paper | N/A | gRNA targeting exon 2 of human RHBDF2 |
| Sequence-based reagent | gRNA targeting exon 2 of human RHBDF2 (GGGTCT CTTTCTCGGGTGGC) | This paper | N/A | gRNA targeting exon 2 of human RHBDF2 |
| Sequence-based reagent | gRNA targeting exon 9 of human RHBDF2 (AAACTC GTCCATGTCATCATCACC) | This paper | N/A | gRNA targeting exon 9 of human RHBDF2 |
| Sequence-based reagent | gRNA targeting exon 9 of human RHBDF2 (ACGGG TGCGATGCCATACGC) | This paper | N/A | gRNA targeting exon 9 of human RHBDF2 |
| Sequence-based reagent | non-targeting siGENOME control pool | Dharmacon | D-001206-13-50 | |
| Sequence-based reagent | siGENOME SMARTpool for human FRMD8 | Dharmacon | M-018955-01-0010 | |
| Sequence-based reagent | siRNA targeting human RHBDF2 (HSS128594) | Thermo Fisher Scientific | Cat#1299001 | |
| Sequence-based reagent | siRNA targeting human RHBDF2 (HSS128595) | Thermo Fisher Scientific | Cat#1299001 | |
| Sequence-based reagent | Human *ACTB* (Hs99999903_m1) | Thermo Fisher Scientific | Cat#4331182 | |
| Sequence-based reagent | Human *ADAM17* (Hs01041915_m1) | Thermo Fisher Scientific | Cat#4331182 | |
| Sequence-based reagent | Human *FRMD8* (Hs00607699_mH) | Thermo Fisher Scientific | Cat#4331182 | |

*Continued on next page*

*Continued*

| Reagent type (species) or resource | Designation | Source or reference | Identifiers | Additional information |
|---|---|---|---|---|
| Sequence-based reagent | Human *RHBDF2* (Hs00226277_m1) | Thermo Fisher Scientific | Cat#4331182 | |
| Sequence-based reagent | Human *TNF* (Hs00174128_m1) | Thermo Fisher Scientific | Cat#4331182 | |
| Commercial assay or kit | BCA Protein Assay Kit | Thermo Fisher Scientific | Cat# 23225 | |
| Commercial assay or kit | Human TNF alpha ELISA Kit | Thermo Fisher Scientific | Cat#88-7346-86 | |
| Commercial assay or kit | SuperScript VILO cDNA synthesis kit | Thermo Fisher Scientific | Cat#11754050 | |
| Chemical compound, drug | 1,10-Phenanthroline | Sigma-Aldrich | Cat#131377–5G | |
| Chemical compound, drug | DSP (dithiobis(succinimidyl propionate)) | Thermos Fisher Scientific | Cat#22585 | |
| Chemical compound, drug | EDTA-free protease inhibitor mix | Roche | Cat#11873580001 | |
| Chemical compound, drug | GW280264X (GW) | (*Lorenzen et al., 2016*); received from Stefan Düsterhöft | N/A | |
| Chemical compound, drug | GI254023X (GI) | (*Lorenzen et al., 2016*); received from Stefan Düsterhöft | N/A | |
| Chemical compound, drug | LPS | Sigma-Aldrich | Cat#L5668-2ML | |
| Chemical compound, drug | nocodazole | Sigma-Aldrich | Cat#M1404 | |
| Chemical compound, drug | PNPP substrate | Thermos Fisher Scientific | Cat#34045 | |
| Chemical compound, drug | Rho kinase inhibitor Y-27632 | Abcam | Cat#ab120129 | |
| Peptide, recombinant protein | Q5 High-Fidelity DNA polymerase | New England Biolabs | Cat#M0491S | |
| Peptide, recombinant protein | Sequencing Grade Trypsin | Promega | Cat#V5111 | |
| Peptide, recombinant protein | HA peptide | Roche | Cat#I2149-.5MG | |
| Peptide, recombinant protein | M-CSF | Gibco | Cat#PHC9501 | |
| Peptide, recombinant protein | IL-3 | Gibco | Cat#PHC0033 | |
| Peptide, recombinant protein | BMP-4 | Invitrogen | Cat#PHC9534 | |
| Peptide, recombinant protein | VEGF | PeproTech | Cat#100–20 | |
| Peptide, recombinant protein | SCF | Miltenyi | Cat#130-094-303 | |
| Peptide, recombinant protein | RhFGF (bFGF) | R and D | Cat#4114-TC | |
| Other | anti-HA magnetic beads | Thermo Fisher Scientific | Cat#88837 | |
| Other | anti-V5 magnetic beads | MBL International | Cat#M167-11 | |
| Other | concanavalin A sepharose | Sigma-Aldrich | Cat#C9017-25ML | |

*Continued on next page*

*Continued*

| Reagent type (species) or resource | Designation | Source or reference | Identifiers | Additional information |
|---|---|---|---|---|
| Other | C18 spin columns | Thermo Fisher Scientific | Cat#89873 | |
| Other | vivaspin concentrator 500 (10,000 kDa MWCO) | Sartorius | Cat#VS0102 | |
| Other | mouse IgG agarose | Sigma-Aldrich | Cat#A0919-5ML | |
| Other | DMEM | Thermo Fischer Scientific | Cat#41965039 | |
| Other | Fetal bovine serum | Thermo Fischer Scientific | Cat#10500064 | |
| Other | Fish skin gelatin | Sigma-Aldrich | Cat#G7765 | |
| Other | KnockOut -DMEM | Thermo Fischer Scientific | Cat#10829 | |
| Other | KnockOut - serum replacement | Thermo Fischer Scientific | Cat#10828 | |
| Other | MEM Non-Essential Amino Acids (100x) | Thermo Fischer Scientific | Cat#11140–035 | |
| Other | GlutaMAX (100x) | Thermo Fischer Scientific | Cat#35050–038 | |
| Other | 2-Mercaptoethanol (1000x) | Thermo Fischer Scientific | Cat#31350–010 | |
| Other | Penicillin-Streptomycin (P/S 100x) | Thermo Fischer Scientific | Cat#15140–122 | |
| Other | 6-well ultra-low attachment plates | Corning | Cat#3471 | |
| Other | X-VIVO 15 | Lonza | Cat#BE04-418 | |
| Other | mTeSR1 | Stemcell Technologies | Cat#12491 | |
| Other | Paraformaldehyde 16% | Electron Microscopy Sciences | Cat#15710 | |
| Other | hESC-qualified Geltrex | Thermo Fischer Scientific | Cat#A1413302 | |
| Other | ProLong Gold antifade reagent with DAPI | Molecular Probes | Cat#P36935 | |
| Software, algorithm | FlowJo (version X 10.0.7r2) | FlowJo, LLC | https://www.flowjo.com/solutions/flowjo | |
| Software, algorithm | Prism (version 7) | GraphPad | https://www.graphpad.com/scientific-software/prism/ | |
| Software, algorithm | MaxQuant (version 1.5.0.35) | (*Cox and Mann, 2008*) | http://www.coxdocs.org/doku.php?id=maxquant:start | |
| Software, algorithm | Perseus (version 1.5.5.3) | (*Tyanova et al., 2016*) | http://www.coxdocs.org/doku.php?id=perseus:start | |
| Software, algorithm | Fiji (version 2.0.0-rc-43/1.52a) | (*Schindelin et al., 2012*) | https://fiji.sc/ | |
| Software, algorithm | Clustal Omega | EMBL-EBI | https://www.ebi.ac.uk/Tools/msa/clustalo/ | |

## Molecular cloning

Human UNC93B1, human iRhom2$^{WT}$, iRhom2$^{\Delta 100}$, iRhom2$^{\Delta 200}$, iRhom2$^{\Delta 300}$, iRhom2$^{\Delta 382}$ iRhom2$^{\Delta 201-300}$, and FRMD8-iRhom2$^{\Delta 300}$ were amplified from human UNC93B1 (BC025669.1), human iRhom2 cDNA (NM_024599.2; Origene (SC122961)) and human FRMD8 cDNA (NM_031904; Addgene (SC107202)) by PCR and cloned with an C-terminal 3xHA tag into the lentiviral vector pLEX.puro using Gibson assembly (New England Biolabs) following the manufacturer's instructions. C-terminal V5-tagged FRMD8 (FRMD8-V5) was amplified from human FRMD8 cDNA (Addgene (SC107202)) by

PCR and cloned into pcDNA3.1(+) using Gibson assembly. All constructs were verified by Sanger sequencing (Source Bioscience, Oxford, UK). pM6P.blast plasmids expressing mouse iRhom2$^{WT}$, iRhom2$^{\Delta268}$ (iRhom2 cub), and iRhom2$^{pDEAD}$ were described previously (*Grieve et al., 2017*).

## Transfection and transduction of cell lines

Human embryonic kidney (HEK) 293T cells were cultured in DMEM (Sigma-Aldrich) supplemented with 10% fetal calf serum (FCS) and 1x penicillin-streptomycin (PS) (all Gibco) at 37°C with 5% CO$_2$. Cells were transiently transfected with DNA using FuGENE HD (Promega). Per 10 cm$^2$ growth area 4 µl FuGENE HD was added to 1 µg DNA diluted in OptiMEM (Gibco). The transfection mix was incubated for 20 min at room temperature and added to cells. Protein expression was analysed 48–72 hr after transfection. For knockdown experiments, siRNA was transfected using Lipofectamin RNAiMax (Invitrogen) following the manufacturer's instructions. Per 6 well 50 pmol of FRMD8 SMARTpool siRNA (Dharmacon; siGENOME Human FRMD8 (83786) siRNA; M-018955-01-0010), non-targeting siRNA control pools (Dharmacon; siGENOME D-001206-13-50), RHBDF2 siRNA (Thermo Fisher Scientific; HSS128594 and HSS128595) were used. Protein expression was analysed 72 hr after transfection.

HEK293T wild-type cell lines stably expressing human UNC93B1-3xHA or human iRhom2-3xHA, and HEK293T iRhom1/2 DKO cell lines expressing iRhom2$^{WT}$, iRhom2$^{\Delta300}$, iRhom2$^{\Delta201-300}$, or FRMD8-iRhom2$^{\Delta300}$ were generated by lentiviral transduction using the pLEX.puro vector as described previously (*Adrain et al., 2012*). Cells were selected by adding 2.5 µg/ml puromycin (Gibco).

## CRISPR/Cas9 genome editing in HEK293T cells

For CRISPR/Cas9-mediated knockout of FRMD8 the plasmid pSpCas9(BB)−2A-Puro (pX459; Addgene plasmid #48139) co-expressing the wild-type *Streptococcus pyogenes* Cas9 and the guide RNA (gRNA) was used. For gRNA design target sequences with a low chance of off targets were selected using online tools (http://crispr.mit.edu; http://www.sanger.ac.uk/htgt/wge). A gRNA targeting exon 7 (ACCCATAAAACGGCAGCTCG), which is present in all FRMD8 isoforms, was cloned into pX459. 1 µg plasmid was transfected into a 6-well of HEK293T cells. Cells were selected with puromycin 48 hr after transfection to eliminate non-transfected cells. Single colonies were selected to establish clonal cell lines. Loss of FRMD8 expression was analysed by western blot and quantitative PCR.

HEK293T iRhom1/2 double-knockout cells were generated using the plasmid pSpCas9(BB)−2A-Puro V2.0 (pX462 V2.0) co-expressing the *S. pyogenes* Cas9 nickase mutant D10A and a guide gRNA. gRNAs targeting exon 3 (GGAACCATGAGTGAGGCCCC, GGGTGGCTTCTTGCGCTGCC) and exon 10 (AGCCGTGTGCATCTATGGCC, CCGTCTCATGCTGCGAGAAC) of *RHBDF1*, and exon 2 (GCAGAGCCGGAAGCCACCCC, GGGTCTCTTTCTCGGGTGGC) and exon 9 (AAACTCGTCCATG TCATCATCACC, ACGGGTGCGATGCCATACGC) of *RHBDF2* were individually cloned into pX462 V2.0. 250 ng of each plasmid were transfected together into a 6-well of HEK293T cells (eight plasmids in total per well). Cells were selected with puromycin 48 hr after transfection and single colonies were selected to establish clonal cell lines. Loss of iRhom1 and iRhom2 was analysed by PCR.

To generate a knock-in of a triple HA tag at the C-terminus of endogenous iRhom2, a homology construct consisting of the triple HA tag (3xHA) flanked at both sides with homology arms of approximately 800 bp was cloned into pcDNA3.1(+). The *RHBDF2* locus was targeted in exon 19 in close proximity to the stop codon using a gRNA (AGCGGTCAGTGCAGCACCT or CAGCGGTCAG TGCAGCACC) cloned into vector epX459(1.1) (generated by subcloning enhanced Cas9 (eSpCas9) v1.1 into plasmid pX459; a kind gift from Dr Joey Riepsaame, University of Oxford). HEK293T cells were treated with 200 ng/ml nocodazole (Sigma-Aldrich) for 17 hr and then transfected with epX459 (1.1) and the pcDNA3.1(+) homology plasmid (0.5 µg each per 6-well). After puromycin selection and single cell cloning, cell clones were tested for the insertion of the 3xHA tag by PCR.

HEK293 ADAM17 knockout cells were kindly provided by Dr Stefan Düsterhöft and have been published previously (*Riethmueller et al., 2016*).

## Mass spectrometry and data analysis

HEK293T cells expressing human UNC93B1-3xHA (control) and human iRhom2-3xHA were used for anti-HA co-immunoprecipitation and analysed by mass spectrometry as described previously

(*Grieve et al., 2017*). Peptides were injected into a nano-flow reversed-phase liquid chromatography coupled to Q Exactive Hybrid Quadrupole-Orbitrap mass spectrometer (Thermo Scientific). The raw data files generated were processed using the MaxQuant (version 1.5.0.35) software, integrated with the Andromeda search engine as described previously (*Cox and Mann, 2008*; *Cox et al., 2011*). Differential protein abundance analysis was performed with Perseus (version 1.5.5.3). A two-sample t-test was used to assess the statistical significance of protein abundance fold-changes. P-values were adjusted for multiple hypothesis testing with the Benjamini-Hochberg correction (*Hochberg and Benjamini, 1990*).

## Co-immunoprecipitation

Cells were washed with ice-cold PBS and then lysed on ice in Trition X-100 lysis buffer (1% Triton X-100, 150 mM NaCl, 50 mM Tris-HCl pH 7.5) supplemented with EDTA-free protease inhibitor mix (Roche) and 10 mM 1,10-Phenanthroline (Sigma-Aldrich). Cell debris were pelleted by centrifugation at 20,000 g at 4°C for 10 min. Proteins were immunoprecipitated by incubation with anti-HA magnetic beads (Thermo Scientific) or anti-V5 magnetic beads (MBL International) for 1 hr at 4°C. Beads were washed with Trition X-100 wash buffer (1% Triton X-100, 300 mM NaCl, 50 mM Tris-HCl pH 7.5). Proteins were eluted in 2x LDS buffer (life technologies) supplemented with 50 mM DTT for 10 min at 65°C.

## Concanavalin A enrichment

N-glycosylated proteins were enriched by incubating cells lysates with concanavalin A sepharose (Sigma-Aldrich) at 4°C for at least 3 hr with over-head rotation. Beads were pelleted (2,500 g, 5 min, 4°C) and washed with Triton X-100 wash buffer. Proteins were eluted in 2x LDS buffer supplemented with 50 mM DTT and 50% sucrose for 10 min at 65°C.

## Cycloheximide chase

To access protein stability, HEK293T cells were treated with 100 µg/ml cycloheximide (CHX; Sigma-Aldrich) for 0–8 hr to block protein synthesis. After incubation, cells were washed with ice-cold PBS and then lysed on ice in Trition X-100 lysis buffer supplemented with EDTA-free protease inhibitor mix and 10 mM 1,10-Phenanthroline. Lysates were centrifuged at 20,000 g at 4°C for 10 min and analysed by SDS-PAGE.

## SDS-PAGE and western blotting

Cell lysates were mixed with 4x LDS buffer (life technologies) supplemented with 50 mM DTT and denatured for 10 min at 65°C prior to loading on 4–12% Bis-Tris gradient gels run in MOPS running buffer (both Invitrogen). Proteins were transferred to a polyvinylidene difluoride (PVDF) membrane (Millipore) in transfer buffer (Invitrogen). The membrane was blocked in 5% milk-TBST (150 mM NaCl, 10 mM Tris-HCl pH 7.5, 0.05% Tween 20, 5% dry milk powder) and then incubated with the primary antibody: mouse monoclonal anti-β-actin-HRP (Sigma-Aldrich, A3854, 1:5000), rabbit polyclonal anti-ADAM17 (abcam; ab39162; 1:2000), rabbit polyclonal anti-FRMD8 (abcam; ab169933; 1:500), rat monoclonal anti-HA-HRP (Roche, 11867423001, 1:2000), goat polyclonal anti-V5 (Santa Cruz, sc-83849, 1:2000), mouse monoclonal anti-transferrin receptor 1, (Thermo Fisher Scientific, 13–6800, 1:2000), and rabbit polyclonal anti-iRhom2 ([*Adrain et al., 2012*]; 1:500). After three washing steps with TBST (150 mM NaCl, 10 mM Tris-HCl pH 7.5, 0.05% Tween 20), membranes were incubated with the secondary antibody for 1 hr at room temperature using either goat polyclonal anti rabbit-HRP (Sigma-Aldrich, A9169, 1:20000), mouse monoclonal anti-goat-HRP (Santa Cruz, sc-2354, 1:5000) or goat polyclonal anti-mouse-HRP (Santa Cruz, sc-2055, 1:5000).

## mRNA isolation and quantitative RT-PCR

Cells were harvested in PBS and pelleted at 3000 g, 5 min, 4°C. RNA was isolated using the RNeasy kit (Qiagen) and reverse transcribed using the SuperScript VILO cDNA synthesis kit (Invitrogen). Resulting cDNA was used for quantitative PCR (qPCR) using the TaqMan Gene Expression Master Mix (Applied Biosystems) and the following TaqMan probes (all Thermo Fisher Scientific): human ACTB (Hs99999903_m1), human FRMD8 (Hs00607699_mH), human RHBDF2 (Hs00226277_m1), and human TNFα (Hs00174128_m1). qPCR was performed on a StepOnePlus system (Applied

Biosystems). Gene expression was normalized to ACTB expression and expressed as relative quantities compared to the corresponding wild-type cell line. Error bars indicate the standard derivation of technical replicates.

## Shedding assay

eight $\times$ $10^4$ HEK293T cells were seeded in triplicates per condition into poly-(L)-lysine (PLL; Sigma-Aldrich)-coated 24-well plates and transfected the next day with 30 ng plasmid DNA encoding Alkaline Phosphatase (AP)-conjugated AREG, HB-EGF or TGFα (received from Prof Carl Blobel). 48 hr after transfection, cells were washed with OptiMEM and then incubated with 200 µl phenolred-free OptiMEM (Gibco) containing either 200 nM PMA, the corresponding volume of the solvent (DMSO), or 200 nM PMA and 1 µM GW (synthesized by Iris Biotech (Marktredwitz, Germany) and kindly provided by Dr Stefan Düsterhöft) for 30 min at 37°C. Cell supernatants were collected, the cells were washed in PBS and lysed in 200 µl Trition X-100 lysis buffer. The activity of AP in cell lysates and supernatants was determined by incubating 100 µl AP substrate p-nitrophenyl phosphate (PNPP) (Thermo Scientific) with 100 µl cell lysate or cell supernatant at room temperature followed by the measurement of the absorption at 405 nm. The percentage of AP-conjugated material released from each well was calculated by dividing the signal from the supernatant by the sum of the signal from lysate and supernatant. The data was expressed as mean of at least three independent experiments, each of which contained three biological replicates per condition.

## Deglycosylation assay

Cells were lysed in Triton X-100 lysis buffer as described above. Lysates were first denatured with Glycoprotein Denaturing Buffer (New England Biolabs) at 65°C for 15 min and then treated with endoglycosidase H (Endo H) or peptide:N-glycosidase F (PNGase F) following the manufacturer's instructions (New England Biolabs).

## Flow cytometry

For ADAM10 and ADAM17 cell surface staining, HEK293T cells were stimulated with 200 nM PMA for 5 min before harvest in PBS. $0.5 \times 10^6$ HEK293T cells were washed with ice-cold FACS buffer (0.25% BSA, 0.1% sodium azide in PBS) and stained with rabbit polyclonal anti-HA antibody (Santa Cruz, sc-805; 0.5 µg diluted in FACS buffer), mouse monoclonal anti-ADAM10 (Biolegend, 352702; 4 µg diluted in FACS buffer) or mouse monoclonal anti-ADAM17 (A300E antibody (*Yamamoto et al., 2012*), kindly provided by Dr Stefan Düsterhöft; 8 µg diluted in FACS buffer) on ice for 45 min. After two washes with FACS buffer, the cells were incubated with Alexa Fluor 488-coupled secondary antibody (Invitrogen, A21202 or A21206); 1:1000 dilution in FACS buffer) on ice for 30 min. Cells were washed twice with ice-cold FACS buffer and then analysed with a BD FACSCalibur (BD Biosciences) and FlowJo software. Cells stained only with the secondary antibody or anti-HA negative cells served as control.

## Immunofluorescence staining and confocal microscopy

HEK293T iRhom1/2 DKO cells ($1.5 \times 10^5$) transduced with indicated iRhom2 constructs were plated on 13 mm PLL-coated glass coverslips in 12-well dishes. In FRMD8-V5 or TACE-V5 overexpression experiments, cells were transfected with 200 ng vector and grown for 72 hr prior to fixation. As indicated, cells were treated with 100 nM bafilomycin for 16 hr before fixation, to inhibit lysosomal degradation. Cells were washed three times in PBS at room temperature and fixed with 4% paraformaldehyde in PBS at room temperature for 20 mins. Fixative was quenched with 50 mM NH$_4$Cl for 5 min. Cells were permeabilised in 0.2% Triton X-100 in PBS for 30 min and epitopes blocked with 1% fish-skin gelatin (Sigma-Aldrich) in PBS for 1 hr. Coverslips were then incubated at room temperature for 2 hr with rabbit anti-HA tag (Cell Signalling, 3724) and goat anti-V5 probe (Santa Cruz, sc-83849) in 1% fish-skin gelatin/PBS. After three PBS washes, coverslips were incubated with Alexa Fluor-coupled secondary antibodies raised in donkey (Invitrogen) for 45 min at room temperature. Cells were subsequently washed three times with PBS and once with H$_2$O, prior to mounting on glass slides with mounting medium containing DAPI (ProLong Gold; ThermoFisher Scientific). Images were acquired with a laser scanning confocal microscope (Fluoview FV1000; Olympus) with a $60 \times 1.4$ NA oil objective and processed using Fiji (ImageJ).

## Culture of human iPSCs

To generate iPSC-derived FRMD8 knockout macrophages, the human iPSC line AH017-13 was used. The AH017-13 line was derived from dermal fibroblasts of healthy donor in the James Martin Stem Cell Facility, University of Oxford as published previously (*Fernandes et al., 2016*). Donors had given signed informed consent for the derivation of human iPSC lines from skin biopsies and SNP analysis (Ethics Committee: National Health Service, Health Research Authority, NRES Committee South Central, Berkshire, UK (REC 10/H0505/71)). AH017-13 iPSCs were cultured feeder cell-free in mTeSR1 (STEMCELL Technologies) on hESC-qualified geltrex (Gibco). iPSCs were fed daily and routinely passaged with 0.5 mM EDTA, or when required using TrypLE (Gibco) and plated in media containing 10 µmol/l Rho-kinase inhibitor Y-27632 (Abcam).

## Genome editing of iPSCs lines

AH017-13 iPSCs were transfected by electroporation using the Neon Transfection System (Invitrogen). $3 \times 10^6$ AH017-13 iPSCs were electroporated (1400 mV, 20 ms, one pulse) in a 100 µl tip with 15 µg pX459-FRMD8-exon7 plasmid DNA (endotoxin-free quality), then plated at a density of $4 \times 10^5$ cells/cm$^2$ and selected 48 hr after transfection with 0.25 µg/ml puromycin. After 48 hr of selection, surviving cells were plated on a feeder-layer of $4 \times 10^6$ irradiated mouse embryonic fibroblasts (MEFs) in 0.1% gelatin-coated 10 cm culture dishes and cultured in hES medium (KnockOut DMEM, 20% KnockOut serum replacement, 2 mM L-Glutamine, 100 µM nonessential amino acids, 50 µM 2-Mercaptoethanol (all Gibco) and 10 ng/mL basic fibroblastic growth factor (bFGF, R and D)). Colonies were manually selected and grown on geltrex in mTeSR1. Clones were analysed by western blot using the anti-FRMD8 antibody, and PCR followed by Sanger sequencing. For PCR DNA was isolated from iPSCs by incubation in DNA isolation buffer (10 mM Tris-HCl (pH 8), 1 mM EDTA, 25 mM NaCl, 200 µg/ml proteinase K added freshly) at 65°C for 30 min. Proteinase K was inactivated at 95°C for 2 min. PCR using Q5 polymerase was performed according to the manufacturer's instructions (New England Biolabs) using primers FRMD8_fw (tgcagATCCATGACGAGGA) and FRMD8_rev (gtgctcgtgacaagacac). The PCR product was purified and sequenced using the primer FRMD8_exon7_fw (GCCAGAGTCTCTTTGCTG) for Sanger sequencing (Source Bioscience, Oxford).

## Differentiation of iPSCs into macrophages

AH017-13 wild-type and FRMD8 knockout clones were analysed by Illumina HumanOmniExpress24 single nucleotide polymorphism (SNP) array at the Wellcome Trust Centre for Human Genetics at the University of Oxford and assessed using KaryoStudio software to confirm normal karyotypes before differentiation into macrophages. For this study iPSCs were differentiated into embryoid bodies (EBs) by mechanical lifting of iPSC colonies and differentiated into macrophages as described in (*van Wilgenburg et al., 2013*). Briefly, iPSCs were grown on a feeder layer of MEFs in hES medium. A dense 10 cm$^2$ well of iPSCs was scored into $10 \times 10$ sections using a plastic pipette tip. The resulting 100 patches were lifted with a cell scraper and cell clumps were transferred into a 6-well ultra-low adherence plate (Corning) containing EB formation medium (hES medium supplemented with 50 ng/ml BMP4 (Invitrogen), 50 ng/ml VEGF (Peprotech) and 20 ng/ml SCF (Miltenyi)) to form EBs. A 50% medium change was performed every second day. On day 5 EBs were harvested. Approximately 60–80 EBs were transferred into a T75 flask containing factory medium (X-VIVO 15 (Lonza) supplemented with 2 mM L-Glutamine, 50 µM 2-Mercaptoethanol, 100 ng/ml M-CSF and 25 ng/mL IL-3, 100 U/ml penicillin and 100 µg/ml streptomycin (all Gibco)). The EBs were fed weekly with fresh factory medium. After approximately two weeks EBs started to produce non-adherent macrophage precursors, which were harvested from the supernatant of EB cultures through a 70 µM cell strainer. Cells were differentiated into mature adherent macrophages for 7 days in macrophage medium (X-VIVO 15 supplemented with 2 mM L-Glutamine, 100 ng/ml M-CSF, 100 U/ml penicillin and 100 µg/ml streptomycin).

## ELISA

iPSC-derived macrophages were harvested from EB cultures, counted and seeded at 25,000 cells per well into 96-well tissue culture plates in triplicates per condition. Macrophages were cultured in macrophage differentiation medium for 7 days, and then activated with 50 ng/ml LPS (Sigma-

Aldrich) in fresh macrophage differentiation medium for 4 hr. For inhibitor treatments cells were incubated with 50 ng/ml LPS and 3 µM GW or GI (synthesized by Iris Biotech (Marktredwitz, Germany) and kindly provided by Dr Stefan Düsterhöft) for 4 hr. Cell culture supernatants were collected and cleared from cells by centrifugation. TNFα in supernatants was measured by ELISA (Human TNF alpha ELISA Ready-SET-Go, eBioscience (88-7346-86)) according to the manufacturer's instructions. Macrophages were lysed in Trition X-100 lysis buffer and protein concentration was determined using a BCA assay (Thermo Scientific). The amount of TNFα in the supernatant was normalised to the protein concentration of the corresponding cell lysate to adjust for differences in TNFα release due to cell numbers.

## Mouse work

Commercially available $Frmd8^{-/-}$ mouse ES cells from KOMP Repository at UC Davis were used to generate $Frmd8^{-/-}$ mice. The mouse ES cells (C57BL/6NTac strain) were injected into blastocysts of Balb/c mice. Chimeras were bred to C57BL/6 to generate $Frmd8^{+/-}$ mice that were used for breeding of the colony and the generation of $Frmd8^{-/-}$ mice. For mice described in *Figure 9—figure supplement 2B*, we excised the LoxP-flanked neomycin resistance gene by breeding $Frmd8^{-/-}$ mice with homozygous Sox2-Cre deleter strain mice. The mouse work was performed under project licenses 80/2584 and 30/2306. Mouse tissues were collected from sacrificed animals and stored on dry ice or at −80°C. Tissues were lysed in Triton X-100 RIPA buffer (1% Triton X-100, 150 mM NaCl, 50 mM Tris-HCl (pH 7.5), 0.1% SDS, 0.5% sodium deoxycholate) supplemented with EDTA-free protease inhibitor mix and 10 mM 1,10-Phenanthroline using a tissue homogeniser (Omni International). Lysates were cleared from cell debris by centrifugation (20,000 g, 4°C, 10 min). Protein concentrations of tissue lysates were determined using a BCA assay.

## Statistical analysis and data presentation

Values are expressed as means of at least three independent experiments with error bars representing the standard deviation. Unpaired, two-tailed t-tests were used for statistical analysis. Shedding assays and ELISA data was analysed using a Mann-Whitney test. Flow cytometry blots shown represent one from at least three experiments with similar outcome.

## Ethics statement

Human iPSC lines were derived from dermal fibroblasts of donors that had given signed informed consent for the derivation of human iPSC lines from skin biopsies and SNP analysis (Ethics Committee: National Health Service, Health Research Authority, NRES Committee South Central, Berkshire, UK (REC 10/H0505/71)).

All procedures on mice were conducted in accordance with the UK Scientific Procedures Act (1986) under a project license (PPL) authorized by the UK Home Office Animal Procedures Committee, project licenses 80/2584 and 30/2306, and approved by the Sir William Dunn School of Pathology Local Ethical Review Committee.

## Cell lines statement

We used HEK293T cells (RRID: CVCL_0063) for analysis of protein-protein interactions, subcellular localisation and loss-of-function experiments. These cells were used for experiments that provided a strong platform of *in vitro* evidence of a relationship between FRMD8 and iRhoms, prior to the generation of iPSC-derived macrophages and FRMD8 knock-out mice. The HEK293T cell line has been tested negative for mycoplasma contamination.

## Acknowledgements

We gratefully acknowledge the support of Oxford's Advanced Proteomics Facility for our mass spectrometry based proteomic screen and Monika Stegmann for statistical analysis of the results. We also thank Genome Engineering Oxford, specifically Joey Riepsaame and Andrew Bassett, who helped us to design and clone guide RNAs for CRISPR/enhanced Cas9 gene editing. We are thankful for the assistance in animal work from the staff of the mouse facility and for support from Elizabeth Robertson, Jonathan Godwin, Angela Moncada Pazos, and Clémence Levet. Immunofluorescent

microscopy was performed in Oxford's Micron imaging facility. We thank members of the Freeman lab for their extensive support throughout this project and their advice on the manuscript. We also thank Stefan Düsterhöft for providing reagents. This research was supported by the Wellcome Trust to MF (grant number 101035/Z/13/Z). The James Martin Stem Cell Facility has received support from the Wellcome Trust ISSF (121302) and MRC (MC_EX_MR/N50192X/1). UK is supported by the Medical Research Council (award number 1374214) and a Boehringer Ingelheim Fonds PhD fellowship. AG received funding from the European Union's Horizon 2020 research and innovation programme under the Marie Sklodowska-Curie grant agreement No 659166. BS is supported by the Medical Research Council and a Boehringer Ingelheim Fonds PhD fellowship.

## Additional information

### Competing interests

Matthew Freeman: Reviewing editor, *eLife*. The other authors declare that no competing interests exist.

### Funding

| Funder | Grant reference number | Author |
| --- | --- | --- |
| Wellcome | 101035/Z/13/Z | Matthew Freeman |
| Medical Research Council | 1374214 | Ulrike Künzel |
| Boehringer Ingelheim Fonds | PhD Fellowship | Ulrike Künzel<br>Boris Sieber |
| Horizon 2020 Framework Programme | 659166 | Adam Graham Grieve |
| Medical Research Council | MC_EX_MR/N50192X/1 | Sally A Cowley |
| Wellcome | Oxford Wellcome Institutional Strategic Support Fund 121302 | Sally A Cowley |

The funders had no role in study design, data collection and interpretation, or the decision to submit the work for publication.

### Author contributions

Ulrike Künzel, Conceptualization, Investigation, Methodology, Writing—original draft, Writing—review and editing; Adam Graham Grieve, Conceptualization, Investigation, Writing—review and editing; Yao Meng, Investigation; Boris Sieber, Investigation, Methodology; Sally A Cowley, Conceptualization, Supervision, Funding acquisition, Investigation, Methodology, Writing—original draft, Writing—review and editing; Matthew Freeman, Conceptualization, Supervision, Funding acquisition, Methodology, Writing—original draft, Writing—review and editing

### Author ORCIDs

Ulrike Künzel http://orcid.org/0000-0003-0648-3325
Adam Graham Grieve http://orcid.org/0000-0001-6420-5724
Yao Meng http://orcid.org/0000-0003-0386-6267
Boris Sieber http://orcid.org/0000-0002-8145-3364
Sally A Cowley http://orcid.org/0000-0003-0297-6675
Matthew Freeman http://orcid.org/0000-0003-0410-5451

### Ethics

Animal experimentation: Human iPSC lines were derived from dermal fibroblasts of donors that had given signed informed consent for the derivation of human iPSC lines from skin biopsies and SNP analysis (Ethics Committee: National Health Service, Health Research Authority, NRES Committee South Central, Berkshire, UK (REC 10/H0505/71)). All procedures on mice were conducted in

accordance with the UK Scientific Procedures Act (1986) under a project license (PPL) authorized by the UK Home Office Animal Procedures Committee, project licenses 80/2584 and 30/2306, and approved by the Sir William Dunn School of Pathology Local Ethical Review Committee.

## Decision letter and Author response
Decision letter https://doi.org/10.7554/eLife.35012.023
Author response https://doi.org/10.7554/eLife.35012.024

## Additional files

### Supplementary files
• Transparent reporting form
DOI: https://doi.org/10.7554/eLife.35012.020

### Data availability
All data generated or analysed during this study are included in the manuscript and supporting files.

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
