## [Decision Letter]

Thank you for submitting your article "FRMD8 promotes inflammatory and growth factor signalling by stabilising the iRhom/ADAM17 sheddase complex" for consideration by *eLife*. Your article has been reviewed by 3 peer reviewers, including Christopher G Burd as the Reviewing Editor, and the evaluation has been overseen by Ivan Dikic as the Senior Editor.

The reviewers have discussed the reviews with one another and the Reviewing Editor has drafted this decision to help you prepare a revised submission.

Summary:

In this study, Kunzel et al. identify FRMD8 as a previously unrecognized regulator of ADAM17 TNF shedding activity. It is shown that FRMD8 associates with ADAM17 and iRhom2 to form what appears to be a tripartite complex. Loss of FRMD8 by siRNA or genome editing results in a reduction in the level of mature ADAM17 and this can be rescued by Bafilomycin and NH4Cl. Physiological studies using human iPSC-derived macrophages and mice indicate that physiological ADAM17-mediated shedding is impaired by loss FRMD8. Overall, this is a superb study that will make an important contribution to the understanding of ADAM17 function in inflammatory signaling.

Essential revisions:

1) The reviewers feel that the functional relevance of a ternary complex of FRMD8, iRhom, and ADAM17 has not been adequately demonstrated. The experiments make a strong case showing that interaction between FRMD8 and iRhoms is essential for mutual stabilization of each protein, but the relevance to ADAM17 is less directly shown. It is shown that over-expression of FRMD8 or iRhom leads to an increase in the amount of the other protein, but the effect on ADAM17 level and processing/activation were not addressed. Does the amount of mature ADAM17 increase by overexpression of FRMD8, iRhom, or both together? The reviewers felt that this is important to bolster the nature of and functional relevance of the complex.

2) The conclusion that FRMD8 prevents lysosome-mediated degradation of mature ADAM17 is inferred from the recovery of mature ADAM17 protein level by bafilomycin and NH4Cl, but it is never directly demonstrated that ADAM17 (or iRhom) is delivered to the lysosome. This could be easily addressed by comparing immunofluorescence patterns of ADAM17 and a lysosome resident in treated cells.

3) Please add a statement (1-2 sentences) about the phenotype of the FRMD8 knock-out mice: if the phenotype differs from ADAM17-/- or iRhom2-/- mice it becomes clear that FRMD8 has additional functions.

---

## [Author Response]

1) The reviewers feel that the functional relevance of a ternary complex of FRMD8, iRhom, and ADAM17 has not been adequately demonstrated. The experiments make a strong case showing that interaction between FRMD8 and iRhoms is essential for mutual stabilization of each protein, but the relevance to ADAM17 is less directly shown. It is shown that over-expression of FRMD8 or iRhom leads to an increase in the amount of the other protein, but the effect on ADAM17 level and processing/activation were not addressed. Does the amount of mature ADAM17 increase by overexpression of FRMD8, iRhom, or both together? The reviewers felt that this is important to bolster the nature of and functional relevance of the complex.

We have addressed these important points extensively, demonstrating more clearly the tripartite relationship and hierarchy of recruitment between iRhom2, ADAM17 and FRMD8, and show that the overexpression of FRMD8 promotes the cell surface residence of both iRhom2 and ADAM17. Below, we describe the experiments performed:

- We used our genetic KO cell lines to perform a series of pairwise interaction assays and show that 1) iRhom2 and FRMD8 can bind to each other in ADAM17 KO cells, 2) ADAM17 and iRhom2 bind one another in FRMD8 KO cells, but 3) there is no interaction between ADAM17 and FRMD8 in iRhom1/2 DKO cells. Overall, this shows that the recruitment of FRMD8 to the iRhom2 N-terminus is independent of ADAM17 binding (new Figure 4).

- Moreover, we have now mapped the specific region within iRhom2 for FRMD8 binding (Figure 3C) and found it to be between amino acids 201 and 300 in the N-terminus of iRhom2.

- Using this specific FRMD8 binding mutant (iRhom2^∆201-300^), we investigated the functional relevance of FRMD8 binding to iRhom2 and to ADAM17 – and show that the loss of FRMD8 binding to iRhom2 inhibits ADAM17-dependent shedding and also fails to rescue levels of mature ADAM17 in iRhom1/2 DKO cells (Figure 3D, E).

- In addition, we performed experiments that show the overexpression of FRMD8 leads to increased localisation of iRhom2 at the plasma membrane. This is direct, as an iRhom2 mutant that cannot bind to FRMD8 does not display enhanced localisation at the cell surface (Figure 5A, B).

- Importantly, these FRMD8 overexpression data highlighted that FRMD8 may stabilise iRhom2/ADAM17 at the cell surface. To test this, we directly fused FRMD8 to iRhom2 – and analysed both its localisation and the localisation of ADAM17. In doing so, we found that the localisation of iRhom2 and ADAM17 closely follow one another, in a FRMD8-dependent manner (Figure 5C-E). More specifically, constitutive recruitment of FRMD8 to iRhom2 led to increased cell surface levels of ADAM17. These new data nicely complement our previous data that show reduced cell surface levels of ADAM17 (Figure 2D/9D, E) and iRhom2 (Figure 7A/8B/9D, E) in various FRMD8 KO contexts.

- We have also shown that constitutive recruitment of FRMD8 leads to increased stability of iRhom2; these data were mirrored by the observation that lack of FRMD8 binding decreased iRhom2 stability (Figure 7—figure supplement 1). The functional relevance to ADAM17 is demonstrated by the new data that show the specific lack of FRMD8 binding causes ADAM17 instability (Figure 3E), leading to lysosomal delivery and degradation (Figure 6F, G).

- As requested, we performed shedding assays to analyse the effect of FRMD8 overexpression of the activity of ADAM17. Unexpectedly, the overexpression of FRMD8 led to a decrease in ADAM17-dependent shedding of EGFR ligands, relative to wild-type (see Author response image 1). We have an explanation for this but have decided not to include it in the revised manuscript. Our previous work has shown iRhoms can promote the degradation of EGFR ligands (Zettl et al., 2011, Cell). Our ADAM17 shedding assays rely on the overexpression of AP-tagged EGFR ligands. As FRMD8 increases the total level of iRhoms (shown throughout this manuscript), this will also trigger degradation of the AP-tagged EGFR ligands, confounding this assay. Further experiments are currently underway to address this delicate balance between iRhom-driven EGFR ligand degradation vs. shedding by iRhom2/ADAM17, but this additional complexity needs further study and will also confuse readers, given that the separate degradation function of iRhoms (which we are extensively studying) is much less appreciated.

- Overall, the data described thus far strengthen our conclusion that iRhom2 is a regulatory subunit of ADAM17 – and that FRMD8 regulates ADAM17's cell surface localisation and stability (this conclusion is further strengthened by the additional data submitted in response to reviewer comment #2).

2) The conclusion that FRMD8 prevents lysosome-mediated degradation of mature ADAM17 is inferred from the recovery of mature ADAM17 protein level by bafilomycin and NH4Cl, but it is never directly demonstrated that ADAM17 (or iRhom) is delivered to the lysosome. This could be easily addressed by comparing immunofluorescence patterns of ADAM17 and a lysosome resident in treated cells.

We include new data that clearly demonstrate the delivery of iRhom2/ADAM17 to LAMP1-positive lysosomal structures in the absence of FRMD8 recruitment. These data now form a new Figure 6 that also includes the bafilomycin/NH_4_Cl data.

- First, we have shown that upon treatment with bafilomycin, a substantial pool of wild-type iRhom2 accumulates in lysosomes (Figure 6A, C).

- The lack of FRMD8 recruitment to iRhoms enhances their level in LAMP1-positive lysosomes (Figure 6B, D).

- Under the same conditions, ADAM17 also appears to be delivered to lysosomes (Figure 6F).

- Overall, these data clearly demonstrate enhanced lysosomal delivery of the iRhom/ADAM17 sheddase complex in the absence of FRMD8 recruitment. Complementing this, additional imaging data show FRMD8 recruitment enhances the cell surface localisation of iRhom/ADAM17 (new Figure 5) as described in the response to reviewer comment #1.

3) Please add a statement (1-2 sentences) about the phenotype of the FRMD8 knock-out mice: if the phenotype differs from ADAM17-/- or iRhom2-/- mice it becomes clear that FRMD8 has additional functions.

We have now further described our FRMD8 KO mice in terms of viability, fertility and quantified effects of FRMD8 KO in mice on ADAM17/iRhoms level. The phenotype of FRMD8 KO mice is weaker than the complete loss of ADAM17 or both iRhoms. A detailed physiological and pathological characterisation of FRMD8 KO mice is on-going and we consider it to be beyond the scope of this manuscript. We now compare the FRMD8 KO to the ADAM17ex/ex mouse in the discussion, and comment on their similar phenotypes. If the similarity in phenotype is true, it interestingly places FRMD8 as a potential regulator of colorectal cancer formation.